# *UBE4B*, a *microRNA-9* target gene, promotes autophagy-mediated Tau degradation

Manivannan Subramanian[1,2,7], Seung Jae Hyeon[3,7], Tanuza Das[4], Yoon Seok Suh[1], Yun Kyung Kim [2], Jeong-Soo Lee [1,2], Eun Joo Song [5✉], Hoon Ryu[3✉] & Kweon Yu [1,2,6✉]

The formation of hyperphosphorylated intracellular Tau tangles in the brain is a hallmark of Alzheimer's disease (AD). Tau hyperphosphorylation destabilizes microtubules, promoting neurodegeneration in AD patients. To identify suppressors of *tau*-mediated AD, we perform a screen using a microRNA (miR) library in *Drosophila* and identify the *miR-9* family as suppressors of human *tau* overexpression phenotypes. *CG11070*, a *miR-9a* target gene, and its mammalian orthologue *UBE4B*, an E3/E4 ubiquitin ligase, alleviate eye neurodegeneration, synaptic bouton defects, and crawling phenotypes in *Drosophila* human *tau* overexpression models. Total and phosphorylated Tau levels also decrease upon *CG11070* or *UBE4B* overexpression. In mammalian neuroblastoma cells, overexpression of *UBE4B* and *STUB1*, which encodes the E3 ligase CHIP, increases the ubiquitination and degradation of Tau. In the Tau-BiFC mouse model, *UBE4B* and *STUB*1 overexpression also increase oligomeric Tau degradation. Inhibitor assays of the autophagy and proteasome systems reveal that the autophagy-lysosome system is the major pathway for Tau degradation in this context. These results demonstrate that UBE4B, a *miR-9* target gene, promotes autophagy-mediated Tau degradation together with STUB1, and is thus an innovative therapeutic approach for AD.

---

[1] Metabolism and Neurophysiology Research Group, KRIBB, Daejeon, Korea. [2] Convergence Research Center of Dementia, KIST, Seoul, Korea. [3] Center for Neuroscience, Brain Science Institute, KIST, Seoul, Korea. [4] Biomedical Research Institute, KIST, Seoul, Korea. [5] Graduate School of Pharmaceutical Sciences and College of Pharmacy, Ewha Womans University, Seoul, Korea. [6] Department of Functional Genomics, UST, Daejeon, Korea. [7] These authors contributed equally: Manivannan Subramanian, Seung Jae Hyeon. ✉email: esong@ewha.ac.kr; hoonryu@kist.re.kr; kweonyu@kribb.re.kr

Alzheimer's disease (AD) is one of the most common age-related neurodegenerative diseases, and includes the pathological hallmarks of extracellular amyloid plaques by abnormally folded amyloid-β 42 (Aβ-42) and intracellular neurofibrillary tangles (NFTs) by Tau hyperphosphorylation in the brain[1,2]. Tau is an essential soluble intracellular protein that associates with and stabilizes axon microtubules[3]. Hyperphosphorylation disrupts the physiological function of Tau, resulting in microtubule destabilization and formation of intracellular NFTs, potentiating neurodegeneration and memory impairment in AD patients. Neurodegenerative disorders with Tau inclusions are referred to as tauopathies[4].

*Drosophila melanogaster* has emerged as an important model system for investigating the pathology of tauopathies at the cellular and molecular levels. *Drosophila* tauopathy AD models exhibit visible phenotypes such as brain vacuole formation, neuromuscular junction defects, larval and adult locomotor defects, impairment of learning and memory, and reduced lifespan[5,6]. Ectopic expression of human *tau* in *Drosophila* eyes induces a rough eye phenotype, which is widely used to screen for modifiers of Tau pathology[7]. Similarly, tissue-specific ectopic expression of human *tau* or mutant *tau* in the *Drosophila* brain or mushroom body induces neurodegeneration, manifesting in phenotypes such as locomotor and cognitive impairment[8]. The pathogenic mechanisms of Tau are similar in humans and *Drosophila*[9,10].

MicroRNAs (miRNAs) are non-coding RNAs comprised of 19 to 25 nucleotides, which suppress target mRNAs by pairing to complementary sequences in the 3' UTRs of target genes, impairing translation and in some cases mRNA stability[11,12]. miRNAs are implicated in diverse brain functions including development, cognition, and synaptic plasticity[13]. Dysregulation of miRNAs can be detrimental, and is associated with several human diseases ranging from metabolic and inflammatory disease to neoplasia[14–16]. miRNA expression-profiling studies have identified that multiple miRNAs are dysregulated in the brains of AD patients, but the functional implications of these changes remain unclear[17].

The ubiquitin-proteasome system (UPS) and autophagy-lysosome system (ALS) are the primary protein degradation pathways in eukaryotic cells[18]. Proteins that undergo proteasomal degradation are polyubiquitinated at Lys48 by ubiquitin ligases and targeted for 26S proteasome complex degradation and proteins poly-ubiquitinated at Lys63 are degraded by the ALS. Interestingly, proteostasis is disrupted in AD brains[19]. A prior study reported that Tau phosphorylation induces ubiquitination and subsequent degradation by the UPS[20]. STUB1 (STIP1 homology and U-Box containing protein1) is an E3 ubiquitin ligase that ubiquitinates phosphorylated Tau for proteasomal degradation in vitro[21] and has also been implicated in the clearance of truncated Tau by the ALS[22].

In the present study, we report the results of a *Drosophila* miRNA library screening, which identified the evolutionarily conserved *miR-9* and its target *CG11070* as strong modifiers of the neurodegenerative *Drosophila* rough eye phenotype induced by human *tau* overexpression. Further, *Drosophila* CG11070 and its mammalian orthologue UBE4B (Ubiquitin conjugation E4 B) alleviated the neurodegenerative phenotypes induced by neuronal human *tau* overexpression in *Drosophila*. In addition, total and phosphorylated Tau in aged *tau*-overexpressing flies was decreased by *CG11070* or mammalian *UBE4B* overexpression. In mammalian neuroblastoma cells and a mouse *tau* overexpression model, UBE4B, and STUB1 E3 ligases ubiquitinated Tau proteins and induced autophagy-mediated Tau degradation. These findings suggest that CG11070/UBE4B is a E4 ubiquitin ligase that regulates Tau degradation via the ALS, and is a putative therapeutic target for the treatment of tauopathies such as AD.

## Results

**Identification of *miR-9* as a modifier of *hTau* in *Drosophila* by genome-wide miRNA screening.** The overexpression of human *tau* (*hTau*) in *Drosophila* eyes using the eye-specific *GMR-GAL4* promoter induces eye neurodegeneration, which is grossly characterized by decreased eye size in the rough eye phenotype[7]. This phenotype was used to screen 131 *UAS-miRNA* lines covering 144 *Drosophila* miRNAs[23]. We analyzed the eye morphology of each line, quantitatively measured eye sizes, and arranged the library in the order of increasing eye size (Fig. 1a, Supplementary Fig. 1a, b). Eye-specific overexpression of many miRNAs in *hTau* flies (*GMR > hTau*) affected the eye size. Quantification of eye sizes was plotted in a volcano plot by the ratio of eye size to *p*-value of miRNA-overexpressing lines relative to the control *GMR > hTau* line (Fig. 1b). The most significant reductions in eye sizes were induced by the overexpression of *miR-9a*, *miR-9b*, and *miR-9c* (Fig. 1c, d), which are members of the evolutionarily conserved *miR-9* family (Fig. 1e). The reduced eye size of *miR-9a*, *miR-9b* or *miR-9c* in the absence of *hTau* expression (Fig. 1c, d, and Supplementary Fig. 1) may be due to the involvement of these *miRNAs* during eye development where these miRNAs may regulate other target genes and affect morphology and development of eye sizes. However, when these *miRNAs* were expressed in the presence of *hTau*, the reduction of eye size was enhanced drastically. This indicated an important regulatory role for the *miR-9* family in *Drosophila* Tau toxicity. We focused on *miR-9a* for subsequent studies, as *miR-9a* has 100% homology to mammalian *miR-9* (Fig. 1e).

**Modulation of hTau by *CG11070*, a *miR-9a* target identified by secondary screening.** Because *miR-9a* was identified as a modifier of Tau toxicity, we searched *miR-9a* target genes using three separate microRNA target prediction programs, TargetScan (www.targetscan.org), miRanda (www.microRNA.org), and miRbase (www.mirbase.org), detecting 34 *miR-9a* targets common to all three platforms (Supplementary Fig. 2a). We then screened the 34 putative target genes against the eye phenotype of *hTau* overexpression (*GMR > hTau*) using stably expressed RNAi knockdown in the *GMR > hTau* background, and arranged the RNAi lines in order of increasing eye size (Fig. 2a, Supplementary Fig. 2b, c). Volcano plot analysis was conducted by the ratio of eye size to *p*-value of the 34 RNAi lines relative to the control *GMR > hTau* flies. *CG11070-RNAi* flies exhibited the most prominent reduction in eye size (Fig. 2b, Supplementary Fig. 2c). The eye size of *CG11070-RNAi* flies in the *GMR > hTau* background (*GMR > hTau + CG11070-RNAi*) was significantly reduced relative to the eye size of control *GMR > hTau* flies (Fig. 2c, d).

Because microRNAs bind to 3'-UTR region of target mRNAs to suppress translation and/or mRNA stability, we assessed binding of *miR-9a* to *CG11070* using a miRNA–mRNA pull-down assay[23]. Similar to the known *miR-9a/miR-9* targets *senseless*[24] and *sNPFR1*[23], we found that *miR-9a* bound to and enriched *CG11070* mRNA in *Drosophila* S2 cells compared to its control scrambled *miRNA* (Fig. 2e). Consistently, eye-specific overexpression of *miR-9a* with *GMR-GAL4* flies decreased *CG11070* mRNA levels (Fig. 2f). Taken together, these findings suggested that *CG11070* was a legitimate target of *miR-9a*.

**The overexpression of *Drosophila CG11070* and its mammalian orthologue UBE4B alleviated *hTau* phenotypes in *Drosophila*.** CG11070 and UBE4B proteins had 26% amino acid identities and similar functional domains (Supplementary Fig. 3a). Interestingly, the 3'-UTR region of *UBE4B* mRNA also contained *miR-9* binding sequences, similar to the 3'-UTR region of *CG11070* mRNA (Supplementary Fig. 3b). In addition to *miR-9a*, UBE4B is

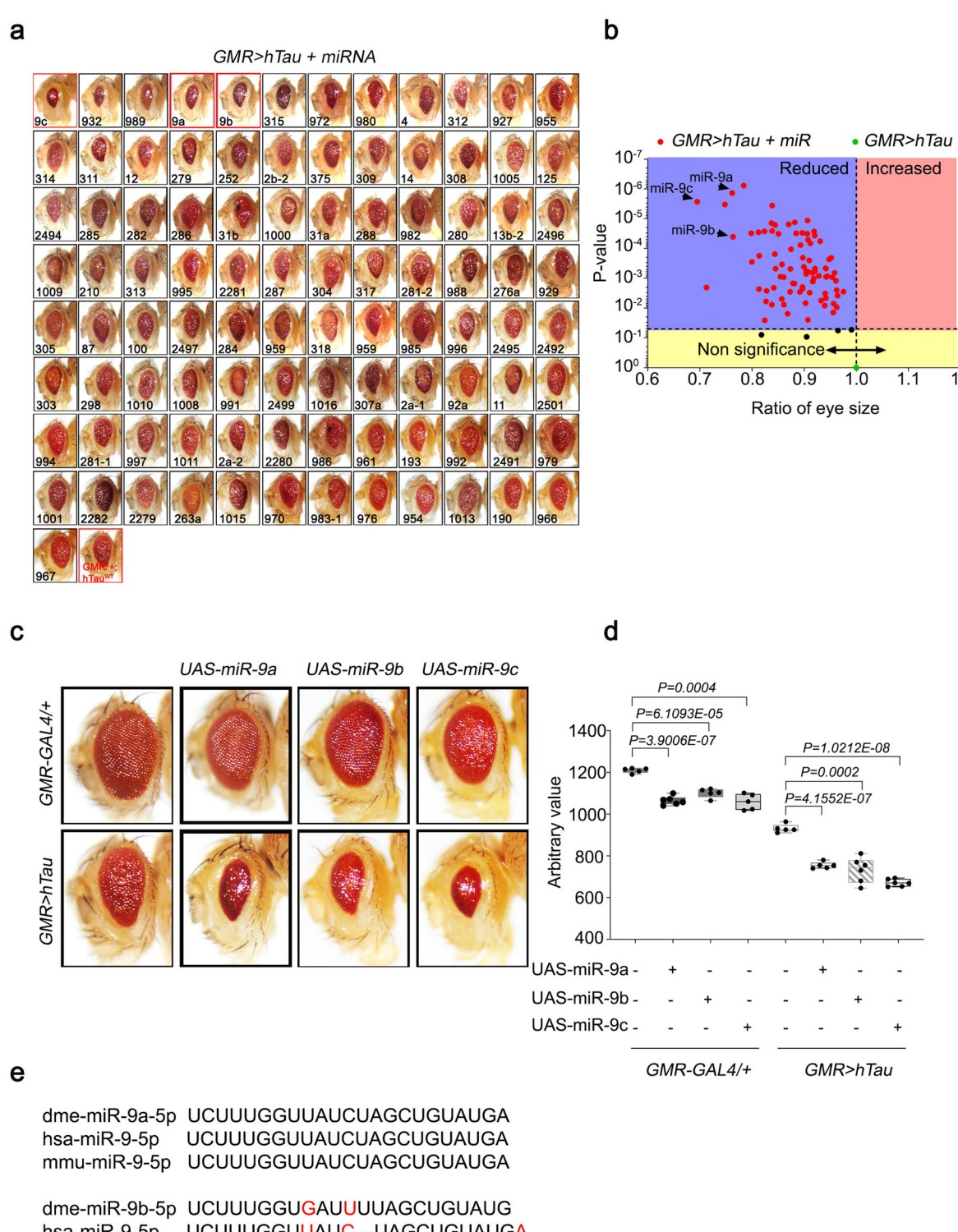

also regulated by *miR-26*, *miR-148/miR-152*, and *miR-15/16/195/424/497*. However, *CG11070* is regulated by *miR-9a* only in *Drosophila*. Therefore, *CG11070/UBE4B* was chosen for further analysis due to similarity in regulation by *miR-9*. Because the knockdown of *CG11070* in *GMR > hTau* flies decreased eye size

relative to control *GMR > hTau* flies, we determined if the over-expression of *CG11070* and its mammalian orthologue *UBE4B* could alleviate *hTau* phenotypes in *GMR > hTau* flies.

In *GMR > hTau* flies, the overexpression of *CG11070* (*GMR > hTau + GC11070*) and *UBE4B* (*GMR > hTau + UBE4B*) increased

**Fig. 1 Genome-wide *Drosophila miRNA* library screening identified *miR-9* family miRNAs as modifiers of *hTau* in *Drosophila* eyes. a** Screening of the *miRNA* library in *GMR* > *hTau Drosophila* eyes revealed significant phenotypic enhancement, as indicated by decreased eye size, in flies overexpressing *miR-9* family miRNAs compared with *GMR* > *hTau* control flies. Eye sizes were arranged from smallest to largest. **b** Volcano plot of mean eye sizes in flies expressing various miRNAs in the *GMR* > *hTau* background versus their respective *p*-values derived from a one-way analysis of variance followed by pairwise *t*-tests and a Bonferroni correction for multiple comparisons. All points above the dotted line can be considered significant. This analysis identified that flies overexpressing *miR-9* family miRNAs exhibited a severe Tau toxicity phenotype, as indicated by decreased eye size. $N = 3$ biologically independent experiments. **c**, **d** The overexpression of *miR-9a*, *miR-9b*, or *miR-9c* in *GMR* > *hTau Drosophila* eyes significantly reduced eye sizes relative to *GMR* > *hTau Drosophila*. $N = 5$ biologically independent experiments. Data are presented as the mean ± s.e.m. Statistical significance was determined with a two-tailed Student's *t*-test. In the box plots the whiskers represent the 5th to 95th percentile range. **e** Alignment of mature *Drosophila miR-9a*, *miR-9b*, and *miR-9c* sequences with human and murine *miR-9* sequences identified that *miR-9a* had 100% identity with mammalian *miR-9* sequences. Statistical source data.

eye size, alleviating the rough eye neurodegenerative phenotype (Fig. 3a, b). Neuron-specific ectopic *hTau* expression (*Elav* > *hTau*) decreased *Drosophila* larva locomotion, which was alleviated by *CG11070* and *UBE4B* overexpression (Fig. 3c, d). The bouton number of neuromuscular junctions (NMJ) in *Drosophila* larvae is directly correlated with locomotion, and is decreased in *Drosophila* AD models[25,26]. Neuronal *hTau*-expressing larvae exhibited significantly reduced bouton numbers relative to the control, which was alleviated by the overexpression of *CG11070* and *UBE4B* (Fig. 3e, f).

The knockdown of *miR-9a* by *miR-9a-sponge* (SP) showed no change in the eye sizes when compared with *GMR* > *hTau* flies (Supplementary Fig. 4a, b). However, the knockdown by *miR-9a-SP* in neurons of *Elav* > *hTau* rescued the larval locomotion phenotype similar to the overexpression of *CG11070* and *hUBE4B* (Supplementary Fig. 4c, d). Similarly, the knockdown by *miR-9a-SP* in neurons also rescued NMJ bouton numbers when compared with the overexpression of *CG11070* and *hUBE4B* in neurons (Supplementary Fig. 4e, f). This data also confirmed that *miR-9a* and *CG11070/UBE4B* forms a common axis involved in regulating Tau toxicity in *Drosophila*.

Because the overexpression of *CG11070* and *UBE4B* alleviated *hTau* phenotypes, we further examined whether the overexpression of these genes affected Tau degradation. We performed western blots on 30-day-old fly heads with eye-specific *hTau* expression (*GMR* > *hTau*), and identified that total hTau protein was significantly decreased by the overexpression of *CG11070* (*GMR* > *hTau* + *GC11070*) and *UBE4B* (*GMR* > *hTau* + *UBE4B*) (Fig. 3g, h). In addition, phosphorylated Tau, detected by AT8 (p-S202/T205) antibody, was also significantly decreased by the overexpression of *CG11070* and *UBE4B* (Fig. 3g, i). Similarly, other Tau phosphorylated forms, detected by AT180 (p-T231) and PHF-1 (p-S396/S404) antibodies, were also reduced in these genotypes (Supplementary Fig. 3c–e). Taken together, these findings suggested that the overexpression of either *Drosophila CG11070* or human *UBE4B* rescued both larval locomotion/NMJ defects and adult eye phenotypes in *hTau*-overexpressing *Drosophila* by increasing Tau degradation.

**Ubiquitination and degradation of Tau by UBE4B and STUB1 in mammalian neuroblastoma cells.** To investigate the mechanism by which UBE4B degraded Tau, we first determined if UBE4B affected Tau ubiquitination in SH-SY5Y neuroblastoma cells (Fig. 4a). Previous studies have demonstrated that UBE4B has E4 ubiquitin ligase activity, and that Tau is ubiquitinated by STUB1 E3 ligase[21,27,28]. STUB1 was chosen due to its role in ubiquitination of Tau proteins and is not regulated by *miR-9*. To confirm the effect of STUB1 on Tau ubiquitination, we evaluated ubiquitination of Tau by STUB1, and found that *STUB1* over-expression alone did not affect Tau ubiquitination (Fig. 4a, lane 3). Similarly, *UBE4B* overexpression alone did not significantly affect Tau ubiquitination (Fig. 4a, lane 4). However, *UBE4B* and

*STUB1* co-expression significantly increased Tau ubiquitination (Fig. 4a, lane 5 compared with lane 3 and 4). Subsequently, we determined that STUB1 activity was important for UBE4B-mediated Tau ubiquitination, as co-expression with a dominant-negative mutant (*STUB1*[H260Q]) failed to enhance Tau ubiquitination (Fig. 4a, lane 6). These results suggested that UBE4B and STUB1 co-regulated Tau ubiquitination.

Due to the essential regulatory roles of these ubiquitin ligases in protein degradation, we examined UBE4B and STUB1-mediated Tau degradation. By inhibiting protein synthesis with cycloheximide (CHX), we observed that Tau degradation was increased by *STUB1* (Fig. 4b, lanes f–i, Fig. 4c) and *UBE4B* overexpression (Fig. 4b, lanes 1–4 and Fig. 4c), and was further increased by co-expression of *STUB1* and *UBE4B* (Fig. 4b, lanes 6–9 relative to lanes 1–4, Fig. 4c). Because *UBE4B* overexpression alone degraded Tau (Fig. 4b, c), we knocked down endogenous *STUB1* with siRNA (Supplementary Fig. 5a) and examined UBE4B-mediated Tau degradation (Supplementary Fig. 5b, c). Interestingly, *UBE4B* overexpression did not affect Tau degradation when *STUB1* was knocked down (Supplementary Fig. 5b, lanes 6–9 compared with lanes 1–4). Similarly, the knockdown of *STUB1* showed no change in Tau degradation when compared with the siControl (Supplementary Fig. 5c). Collectively, these results indicated that UBE4B was a critical factor that enhanced the ubiquitination activity of STUB1 to ubiquitinate and degrade Tau. To evaluate the biochemical interactions of UBE4B with STUB1 and Tau, we performed immunoprecipitation analyses. When *Tau* was co-expressed with *UBE4B* and *STUB1*, UBE4B co-precipitated with STUB1 (Fig. 4d). However, UBE4B did not directly interact with STUB1, as UBE4B did not co-precipitate with STUB1 in the absence of Tau (Fig. 4d). Previous studies demonstrated that STUB1 directly interacts with and ubiquiti-nates Tau, targeting it for degradation[21,27,28]. UBE4B also directly interacted with Tau protein (Fig. 4e). These results suggested that UBE4B did not directly interact with STUB1, but rather that Tau mediated the interaction between STUB1 and UBE4B.

**STUB1 knockdown reduces UBE4B-mediated alleviated hTau phenotypes in Drosophila.** Since the knockdown of *STUB1* in neuroblastoma cell lines showed no change in Tau levels in the presence of *UBE4B* (Supplementary Fig. 5b, c), we further examined the importance of *STUB1* in the in vivo *Drosophila* model system. Studies from the in vivo model system showed that the knockdown of *STUB1* gene in the eyes of flies expressing *GMR* > *hTau* + *hUBE4B*, significantly reduced eye phenotype when compared with *GMR* > *hTau* + *hUBE4B* flies (Fig. 5a, b). Similarly, the knockdown of *STUB1* in neurons expressing *Elav* > *hTau* + *hUBE4B* also significantly reduced larval locomotion phenotype when compared with *Elav* > *hTau* + *hUBE4B* in neurons (Fig. 5c, d). However, NMJ phenotype showed no change in larvae expressing *Elav* > *hTau* + *hUBE4B* + *STUB1-RNAi* when compared with *Elav* > *hTau* + *hUBE4B* larvae (Fig. 5e, f). These

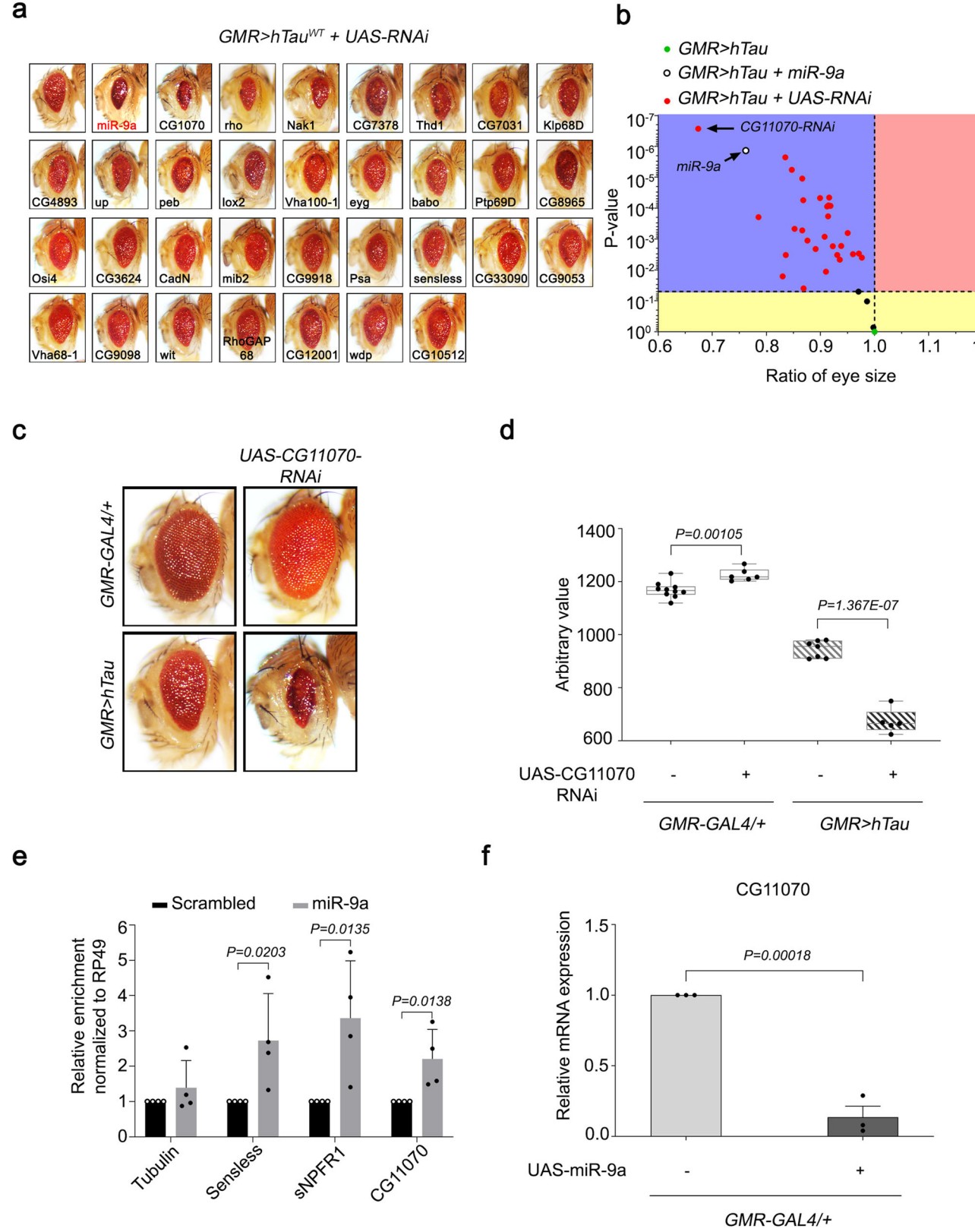

studies indicate that alleviated *hTau* phenotypes by *hUBE4B* overexpressing are dependent on STUB1 function.

**Degradation of Tau by UBE4B and STUB1 in the *Tau-BiFC* mouse model.** To determine whether in vivo overexpression of

*UBE4B* and *STUB1* affects Tau degradation, we generated *AAV-CMV-UBE4B* and *AAV-CMV-STUB1* constructs (Fig. 6a). AAVs were delivered to the dentate gyri of *Tau-BiFC* mice by stereotaxic injection (Fig. 6b, c) and pathological examinations were performed as shown in the work flow (Fig. 6d). In the *Tau-BiFC*

**Fig. 2 CG11070, a miR-9a target identified from secondary screening, strongly modulated hTau in Drosophila eyes. a** Screening of flies with RNAi knockdown of miR-9a targets in GMR > hTau Drosophila eyes identified a significant reduction in eye size in CG11070-RNAi flies relative to the control. **b** Volcano plot of mean eye sizes of flies expressing various miR-9a target gene RNAis in GMR > hTau flies versus their respective p-values derived from a one-way analysis of variance followed by pairwise t-tests and a Bonferroni correction for multiple comparisons. All points above the dotted line can be considered significant. CG11070-RNAi flies exhibited more severe ocular Tau toxicity, as demonstrated by decreased eye size. N = 3 biologically independent experiments. **c**, **d** The knockdown of CG11070 (CG11070-RNAi) in GMR > hTau flies significantly decreased eye size relative to GMR > hTau controls. N = 5 biologically independent experiments. In the box plots the whiskers represent the 5th to 95th percentile range. Data are presented as the mean ± s.e.m. Statistical significance was determined with a two-tailed Student's t-test. **e** miRNA–mRNA–RISC pull-down assays in Drosophila S2 cells revealed that miR-9a bound to CG11070 mRNA. Transfection of miR-9a enriched CG11070 mRNA levels, similar to the known miR-9a targets senseless and sNPFR1, as demonstrated by qRT-PCR. N = 4 biologically independent experiments. Data are presented as the mean ± s.e.m. Statistical significance was determined with a two-tailed Student's t-test. **f** Expression of miR-9a using GMR-GAL4 significantly decreased CG11070 expression. N = 3 biologically independent experiments. Data are presented as the mean ± s.e.m. Statistical significance was determined with a two-tailed Student's t-test. Statistical source data.

mouse model system, Tau oligomerization can be visualized by BiFC fluorescence[29]. Consistent with the in vitro data, the over-expression of either UBE4B or STUB1 induced degradation of Tau oligomer, as indicated by decreased Tau fluorescence relative to control (Fig. 6e). Similarly, co-overexpression of UBE4B and STUB1 in Tau-BiFC mice further decreased Tau fluorescence (Fig. 6e, f). Similar to reduction of oligomeric Tau levels, phosphorylated Tau levels in the dentate gyrus of Tau-BiFC mice, as detected by the AT8 (p-S202/T205) antibody, were also decreased by the overexpression of either UBE4B or STUB1, and were further decreased by co-expression of UBE4B and STUB1 (Fig. 6e and g). Additional phosphorylated forms of Tau were detected by AT180 (p-T231) and PHF-1 (p-S396/S404) antibodies. Both p-T231 and p-S396/S404 Tau levels were also decreased by the overexpression of either UBE4B or STUB1 alone, and were further decreased by co-expression of UBE4B and STUB1 (Supplementary Fig. 6a–c). These results demonstrated that UBE4B and STUB1 additively degraded Tau in vivo.

**UBE4B and STUB1-mediated autophagy Tau degradation.** The two reported major pathways of Tau degradation are the UPS and ALS[30,31]. To determine the primary pathway of UBE4B and STUB1-mediated Tau degradation, we co-overexpressed UBE4B and STUB1 with Tau in neuroblastoma cells, and treated the cells with either the UPS inhibitor MG132 or the ALS inhibitors. Interestingly, MG132 treatment did not inhibit Tau degradation relative to the control cells (Fig. 7a). In contrast, Chloroquine significantly inhibited Tau degradation (Fig. 7b). In addition, pepstatin A (PEPA) alone and E64D plus PEPA (E64D + PEPA), autophagy inhibitors, significantly inhibited Tau degradation (Fig. 7c), suggesting that ALS was the major pathway of UBE4B and STUB1-mediated Tau degradation in neuroblastoma cells. Our in vitro study showed that the turnover of monomeric Tau molecule by UBE4B/STUB1 is preferentially mediated by autophagy-dependent manner rather than the proteasome pathway in SH-SY5Y cells (Fig. 7a–c).

Because autophagy inhibitor treatment blocked autophagic Tau degradation in vitro, we tested this in the in vivo system by injecting autophagy inhibitors into the dentate gyri of Tau-BiFC mice with co-expression of UBE4B and STUB1 (Fig. 7d, e). We measured oligomeric Tau and phosphorylated Tau levels in the dentate gyrus of Tau-BiFC mice detected with the AT8 antibody (S202/T205) (Fig. 7f–h). Inhibition of ALS by Chloroquine and E64D + PEPA significantly increased oligomeric and phosphorylated Tau (S202/T205) (Fig. 7f–h). Similarly, the phosphorylated forms of Tau p-S396/S404 (PHF-1) and p-T231 (AT180) were also significantly increased in Chloroquine and E64D + PEPA-treated Tau-BiFC mice (Supplementary Fig. 7a–d). Since ALS disruption changes LC3 and p62/SQSTM1 levels, which are inversely correlated with the efficiency of autophagy[32], we

measured LC3 and p62 levels in our model system. Co-expression of UBE4B and STUB1 significantly decreased LC3 and p62 levels relative to control in the dentate gyri of Tau-BiFC mice, whereas autophagy inhibitors elevated LC3 and p62 levels relative to the control in the dentate gyri of UBE4B + STUB1-expressed Tau-BiFC mice (Fig. 7i, j). Furthermore, autophagy inhibitors modulated BECN1 levels compared to the control in the dentate gyri of UBE4B + STUB1-expressed Tau-BiFC mice (Supplementary Fig. 8). Collectively, these results suggested that monomeric and oligomeric Tau degradations by STUB1 and UBE4B are mediated through autophagy pathway rather than the ubiquitin-proteasome system.

## Discussion

In the present study, we demonstrated that Drosophila miR-9a regulated Tau toxicity and increased Tau phosphorylation in the eyes. MiR-9a-mediated Tau toxicity arose through disruption of the miR-9a target gene CG11070, the human orthologue of UBE4B, which when overexpressed mimicked the miR-9a knockdown phenotype. The overexpression of either Drosophila CG11070 or human UBE4B alleviated Tau-associated phenotypes, including increased eye size, improved larval crawling/NMJ phenotypes, and decreased total and phosphorylated Tau levels. Further, we demonstrated that clearance of Tau proteins by UBE4B occurred primarily through ubiquitin-dependent ALS rather than the proteasome system.

Previous studies have demonstrated that miRNAs mediate neurodegeneration, including that of AD[33,34]. AD-related miR-NAs regulate multiple stages of AD pathology and enhance Tau toxicity[35,36]. Studies in AD cadaver samples demonstrated that miR-125b is increased in AD, and the overexpression of miR-125b in mice increases Tau hyperphosphorylation by regulating kinases and phosphatases[37]. Also, downregulation of miR-132/212 promotes Tau phosphorylation, which in turn enhances AD phenotypes[38]. In addition to these miRNAs, miR-138 regulates Tau phosphorylation through GSK3β[39] and miR-922 affects Tau phosphorylation through UCHL1[40]. Functional analysis of miR-9 has revealed its involvement in the regulation of neuronal progenitor cells and differentiation of neuronal cells during development[41]. In AD patients, miR-9 expression is elevated[42]. In Drosophila, miR-9a plays an important role in the development of sensory organs by suppressing its target gene senseless[24] and also regulates body growth through sNPFR signaling[23]. In the Drosophila Tau AD model, we observed increased Tau toxicity when miR-9a was overexpressed. The Drosophila miR-9a binding sequence was 100% identical to that of mammalian miR-9 (Fig.1e). Our studies clearly demonstrated that miR-9a regulated Tau toxicity, exacerbating the rough eye phenotype and significantly decreasing eye size.

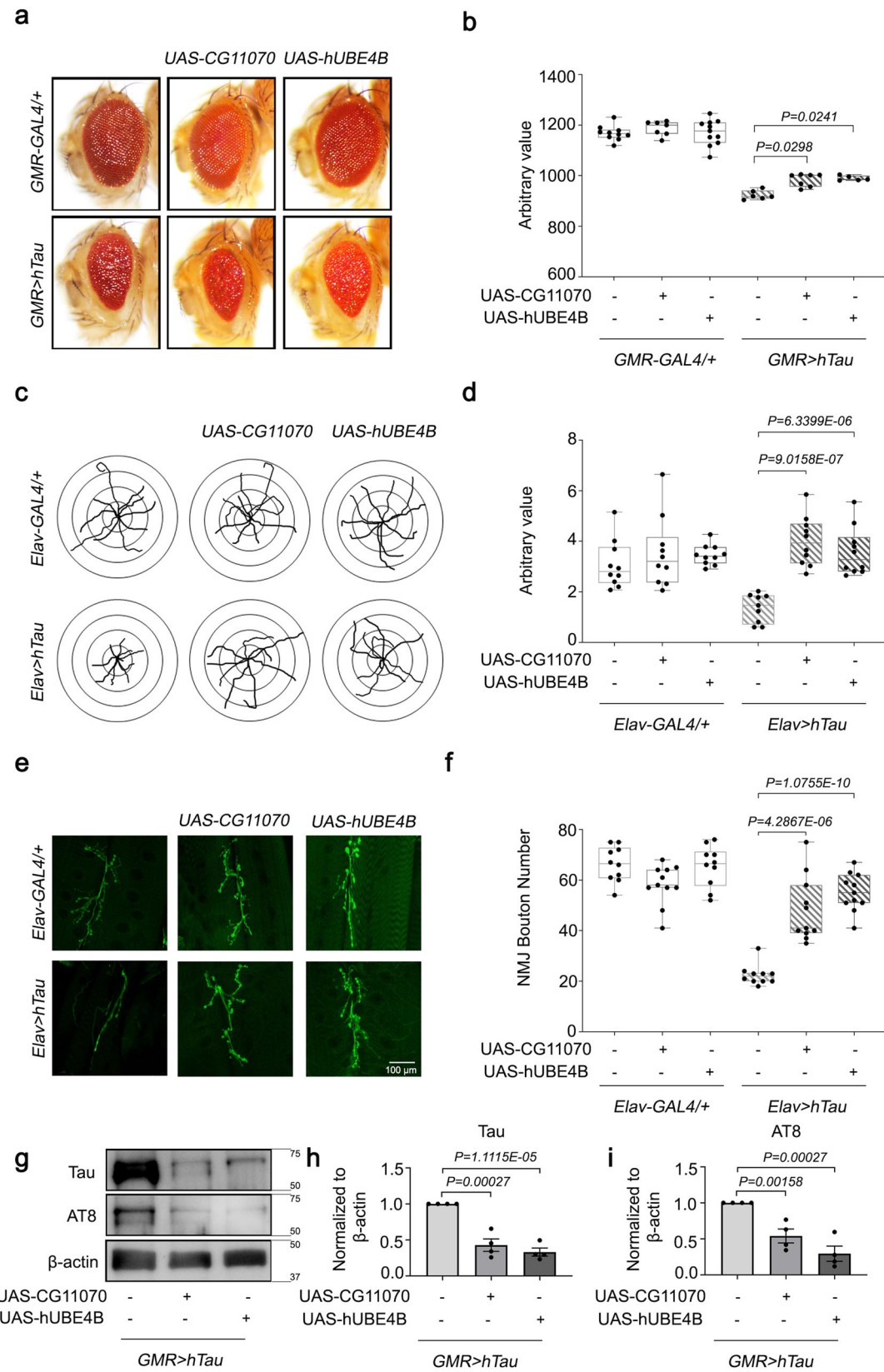

*miR-9a* targets a multitude of genes that are involved in diverse cellular processes, and we used in silico analysis to identify 34 putative targets, which were predicted as *miR-9a* target genes using three separate in silico platforms. We then performed a secondary screening of the putative 34 *miR-9a* target genes to identify a distinct gene, *CG11070*, as a strong modifier of *hTau* (Fig. 2). Mammalian orthologues of *Drosophila CG11070* include *UBE4A* and *UBE4B*. *UBE4B*, but not *UBE4A*, contained *miR-9* binding sequences in its 3′ UTR like those of *CG11070* (Supplementary Fig. 3b), and is extensively expressed in the brain[43]. We

**Fig. 3 The overexpression of *Drosophila CG11070* and its mammalian orthologue *UBE4B* alleviated *hTau* phenotypes in *Drosophila*. a, b** Eye-specific overexpression of *Drosophila CG11070* or its mammalian orthologue *UBE4B* in *GMR > hTau* flies increased eye size relative to *GMR > hTau* controls. N = 5 biologically independent experiments. In the box plots the whiskers represent the 5th to 95th percentile range. **c, d** Neuronal overexpression of *Drosophila CG11070* or its mammalian orthologue *UBE4B* using Elav-Gal4 in *Elav > hTau* flies significantly increased larval crawling. N = 10 biologically independent experiments. In the box plots the whiskers represent the 5th to 95th percentile range. **e, f** Neuronal overexpression of *Drosophila CG11070* or its mammalian orthologue *UBE4B* using Elav-Gal4 in *Elav > hTau* flies significantly increased synaptic bouton numbers relative to *Elav > hTau* controls. Scale bar 100 µm. N = 10 biologically independent experiments. In the box plots the whiskers represent the 5th to 95th percentile range. **g–i** Western blotting revealed that ocular overexpression of *Drosophila CG11070* or its mammalian orthologue *UBE4B* in *GMR > hTau* flies significantly reduced total and phosphorylated Tau protein levels relative to *GMR > hTau* controls. N = 4 biologically independent experiments. Data are presented as the mean ± s.e.m. Statistical significance was determined with a two-tailed Student's *t*-test. Statistical source data.

therefore considered *UBE4B* as the mammalian orthologue of *Drosophila CG11070* in the context of *miR-9a* regulation of Tau toxicity. The overexpression of *UBE4B* or *CG11070* in *Drosophila* not only alleviated *hTau* eye neurodegenerative phenotypes, larval crawling defects, and synaptic dysfunction in NMJs (Fig. 3a, f), but also decreased total and phosphorylated Tau levels (Fig. 3g–i, Supplementary Fig. 3c–e). These results indicated that Tau degradation was strongly regulated by UBE4B/CG11070. Since Tau is involved in maintaining the stability of microtubules in neurons, the degradation of Tau proteins affects microtubule stability as seen in tauopathy and Alzheimer's disease models. However, recent studies have shown that reduction in Tau proteins leads to increased accumulation of MAP6 proteins on the microtubules and enhances the stability of microtubules in neurons[44]. The rescue phenotype seen in *UBE4B/CG1170* may be a result of compromised function of Tau and MAP6 proteins in the neurons. Reduction in Tau proteins leads to reduced pTau levels in *UBE4B/CG11070* rescue animals, due to increased ubiquitination of Tau proteins. Studies have shown that the acetylation of Tau inhibits degradation of phosphorylated Tau and contributes to increased tauopathy[45]. Both ubiquitination and acetylation shares common Lys residue Lys280[46] which is involved in stability of Tau proteins. The UBE4B proteins are distinct from E3 ligases in their U-Box domains, which possess both E3 ligase and E4 ligase activities[47]. UBE4B ligase ubiquitinates the tumor suppressor genes p53, p63 and p73, and Wallerian degeneration proteins in conjunction with various E3 ligases[48]. In the *Drosophila* spinocerebellar ataxia type 3 model, UBE4B ubiquitinates ataxin, targeting it for proteasomal degradation. However, the dominant-negative *UBE4B* mutation enhances neurodegeneration[49]. These previous studies suggest that UBE4B potentiates the degradation of various proteins through the UPS, unlike our current findings, which suggest synergistic regulation of Tau degradation through the ALS in conjunction with STUB1.

Prior reports have demonstrated that Tau is ubiquitinated by two E3 ubiquitin ligases, STUB1[21,27,28] and TRAF6[50], both of which are colocalized with NFTs in AD brains. In the present study, we found that Tau was ubiquitinated by the E3 ligase STUB1 in the presence of UBE4B, a further ubiquitin conjugation factor (Fig. 4a). In addition, Tau degradation was increased by *UBE4B* expression, and further increased by *STUB1* and *UBE4B* co-expression (Fig. 4b, c). Studies on Tau degradation have shown that the proteasomal pathway plays a crucial role. Inhibition of proteasome in HEK cells increases the levels of full-length Tau proteins[50]. Similar results were seen in SH-SY5Y cells with increased accumulation of both full-length Tau protein and mutant P301L Tau protein levels when proteasome degradation is inhibited[51,52]. However, from our studies, STUB1- and UBE4B-mediated Tau degradation was inhibited by the autophagy inhibitors such as Chloroquine, PEPA, and E64D (Fig. 7b, c) but not by the proteasomal inhibitor MG132 (Fig. 7a), suggesting that STUB1/UBE4B-mediated Tau degradation was facilitated by

autophagy rather than the UPS. In FTDP-17 mutant P301S mice, trehalose treatment promotes autophagy with reduced insoluble and pTau proteins[53]. Similarly, the knockout of *Atg7* (autophagy marker) in neurons give rise to increased pTau levels and neurodegeneration with aging[54]. A number of autophagy inhibitors, including chloroquine, $NH_4Cl$, 3-methyl adenine (3-MA), and cathepsin inhibitors, delay Tau degradation and enhance the formation of high molecular weight Tau aggregates[55,56]. Contrastingly, the autophagy inducer rapamycin facilitates insoluble Tau degradation, alleviating Tau toxicity in *Drosophila*[30]. Trehalose, an mTOR-independent autophagy activator[31], improves neuronal survival by decreasing Tau aggregation in tauopathy mouse model[57,58]. This suggests that basal autophagy activity is essential for prevention of neuronal Tau aggregate accumulation, which is regulated in part by UBE4B activation. Considering autophagy inhibitors not only modulate neuronal autophagy pathway but also affect neuroinflammation pathway, it remains to be determined whether other neuroinflammatory pathways are involved in Tau pathology beyond the autophagy pathway in future studies.

Lys63-linked polyubiquitination of Tau facilitates the formation of disease-associated Tau inclusions, which are preferentially cleared by the ALS[59], while Lys48-linked polyubiquitinated Tau is likely degraded by the UPS[60]. Lys63 ubiquitinated substrates are recognized by autophagy receptors present on autophagosomes, mediating ALS-dependent degradation[61–63]. Recent studies have shown that the knockdown of *UBE4B* affects Lys48 and Lys63 polyubiquitination in Tax binding proteins and Tax mediated activation of NF-κB[64]. Therefore, in the context of the present study, UBE4B acted synergistically with STUB1 to facilitate Tau degradation by ALS, and could polyubiquitinate Tau proteins by Lys63 ubiquitin linkage. The specific ubiquitin chain linkage for Tau ubiquitination by STUB1/UBE4B is a potential target for lysosomal targeting of Tau. However, the mechanism underlying ALS-mediated Tau degradation remains to be elucidated in future studies.

Tau oligomers play a central role in tauopathies. To understand the importance of Tau oligomers in vivo, Bimolecular Fluorescence Complementation (BiFC) technology was applied to visualize Tau oligomerization[65]. In this system, full-length human Tau protein is fused to the non-fluorescent N- and C-terminal termini of Venus fluorescent protein. In the transgenic *TauP301L-BiFC* mouse model, Venus fluorescence is activated only when Tau is aggregated[29]. In our studies using the *TauP301L-BiFC* mouse model, the overexpression of *UBE4B*, *STUB1*, or both significantly reduced the fluorescence generated by Tau oligomers (Fig. 6). Similarly, Tau phosphorylation was also decreased in the dentate gyri of *TauP301L-BiFC* mice with the overexpression of *UBE4B*, *STUB1*, or both. These results indicate that Tau oligomers, a primary cause of tauopathy, can be degraded by UBE4B and STUB1 in vivo. In *TauP301L-BiFC* mice, treatment with the autophagy inhibitors (Chloroquine, PEPA, and E64D) increased oligomeric and phosphorylated Tau levels

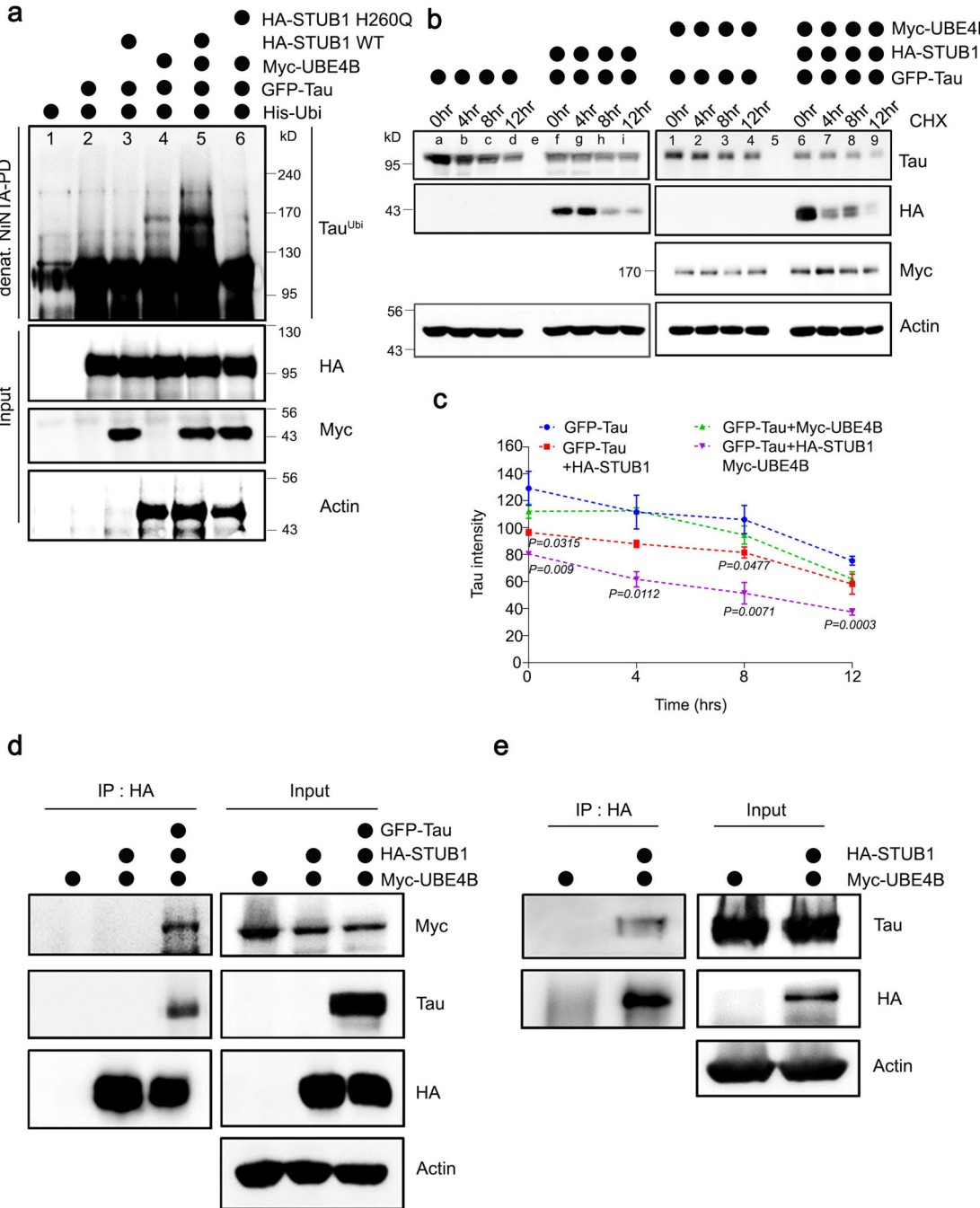

**Fig. 4 Tau was ubiquitinated and degraded by UBE4B and STUB1 in mammalian neuroblastoma cells. a** *Tau* was co-expressed with *His-Ubiquitin*, *UBE4B*, and *STUB1 WT* or dominant-negative mutant *STUB1^H260Q* in SH-SY5Y neuroblastoma cells. Ubiquitinated Tau was significantly increased by co-expression of *UBE4B* and *STUB1* (lane 5), but not by expression of *UBE4B* (lane 4) or *STUB1* (lane 3) alone. Tau ubiquitination by co-expression of *UBE4B* and *STUB1* required the ligase activity of STUB1 (lanes 5 and 6). **b**, **c** Tau protein degradation was enhanced by co-expression of *UBE4B* and *STUB1* (lane 6-9) compared with expression of *UBE4B* (lane 1-4) or *STUB1* (lane f-i) alone. Quantification of Tau levels was normalized to the amount of β-actin protein in each case. Data represent the mean ± s.e.m of three independent experiments (*P < 0.05, **P < 0.005, ***P < 0.001 two-tailed Student's *t*-test). **d** *UBE4B* was co-expressed with *HA-STUB1* and *Tau* in SH-SY5Y cells and immunoprecipitated on anti-HA-agarose beads. UBE4B did not directly interact with STUB1 in the absence of Tau, but indirectly interacted with STUB1 in the presence of Tau. **e** *HA-UBE4B* was co-expressed with *Tau* in SH-SY5Y cells and immunoprecipitated on anti-HA-agarose beads. Co-precipitated Tau was detected by Western blot, revealing that Tau directly interacted with UBE4B. All western blots were performed more than three times. Statistical source data.

(Fig. 7b, c, f–h, Supplementary Fig. 7). Notably, while the overexpression of *UBE4B* and *STUB1* decreased protein levels of LC3, p62, and BECN1, autophagy markers, autophagy inhibitors significantly elevated LC3, p62, and BECN1 levels in the dentate gyri of *AAV-STUB1/UBE4B* injected *TauP301L-BiFC* mice (Fig. 7i–l and Supplementary Fig. 8). Collectively, these results further suggested that UBE4B promoted oligomeric Tau clearance via ALS through a STUB1-dependent mechanism.

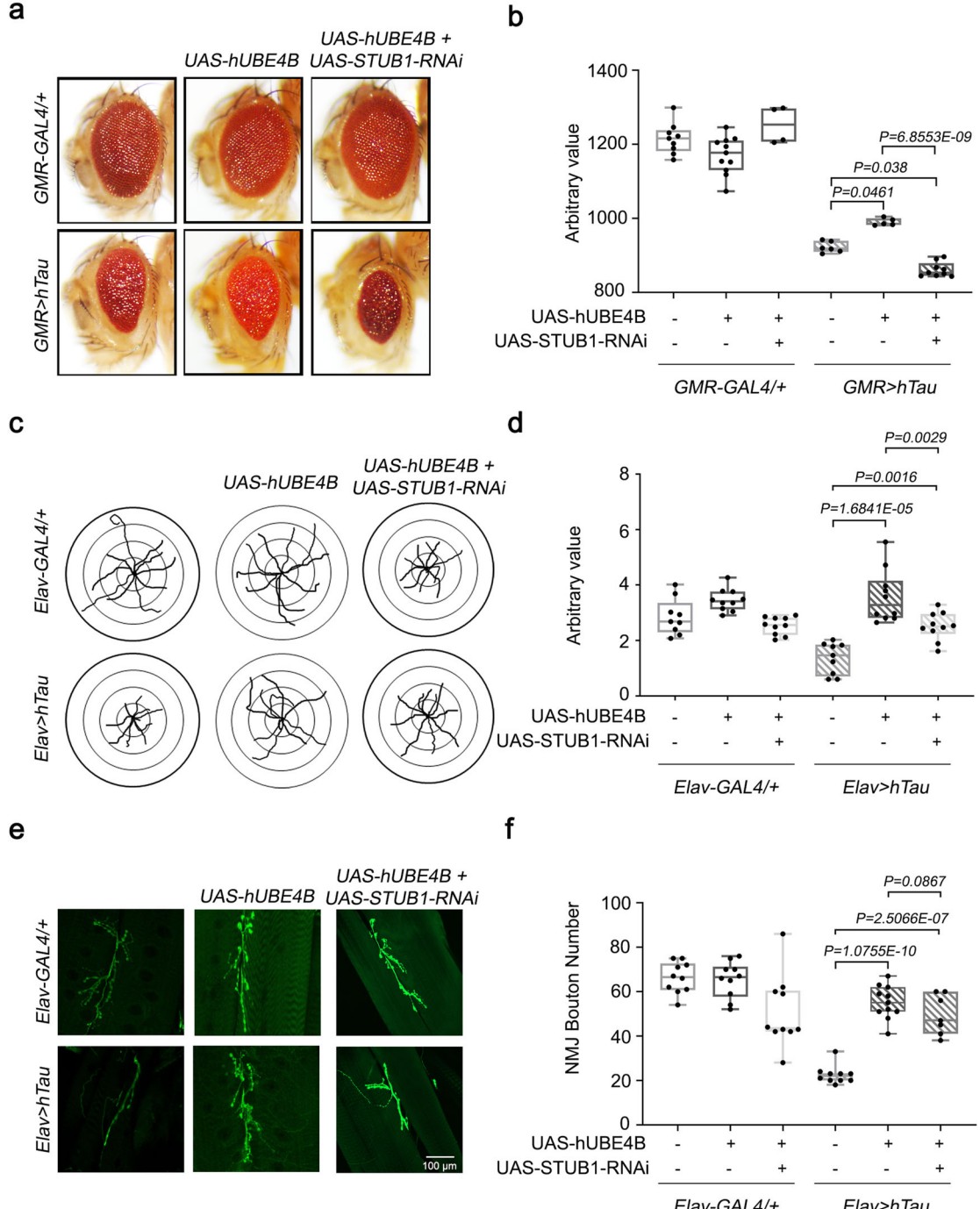

**Fig. 5 *STUB1* knockdown reduces *UBE4B*-mediated alleviated *hTau* phenotypes in *Drosophila*. a, b** Eye-specific knockdown of *Drosophila STUB1* significantly reduced eye size relative to *GMR > hTau + hUBE4B* flies. *N* = 5 biologically independent experiments. In the box plots the whiskers represent the 5th to 95th percentile range. **c, d** Neuronal knockdown of *Drosophila STUB1* using *Elav-Gal4* in *Elav > hTau + hUBE4B* flies significantly reduced larval crawling. *N* = 10 biologically independent experiments. In the box plots the whiskers represent the 5th to 95th percentile range. **e, f** Neuronal knock down of *Drosophila STUB1* using *Elav-Gal4* in *Elav > hTau + hUBE4B* flies significantly reduced synaptic bouton numbers relative to *Elav > hTau* controls. Scale bar 100 μm. *N* = 10 biologically independent experiments. In the box plots the whiskers represent the 5th to 95th percentile range. Data are presented as the mean ± s.e.m. Statistical significance was determined with two-tailed Student's *t*-test. Statistical source data.

In the present study, we identified that *Drosophila CG11070* and its mammalian orthologue *UBE4B*, which are targets of *miR-9a/miR-9*, rescued human Tau phenotypes in flies by decreasing total and phosphorylated Tau levels. In mammalian neuroblastoma cells, UBE4B-mediated Tau clearance was accelerated by co-expression of *STUB1*, which encodes an ubiquitin E3 ligase for Tau. In the dentate gyri of *Tau BiFC*

mice, the overexpression of *UBE4B* and *STUB1* also decreased oligomeric Tau and phosphorylated Tau levels. These Tau degradations occurred primarily via ALS in mammalian in vitro and in vivo systems. These results demonstrated that UBE4B promoted autophagy-mediated Tau degradation synergistically with STUB1, providing an innovative therapeutic approach for AD.

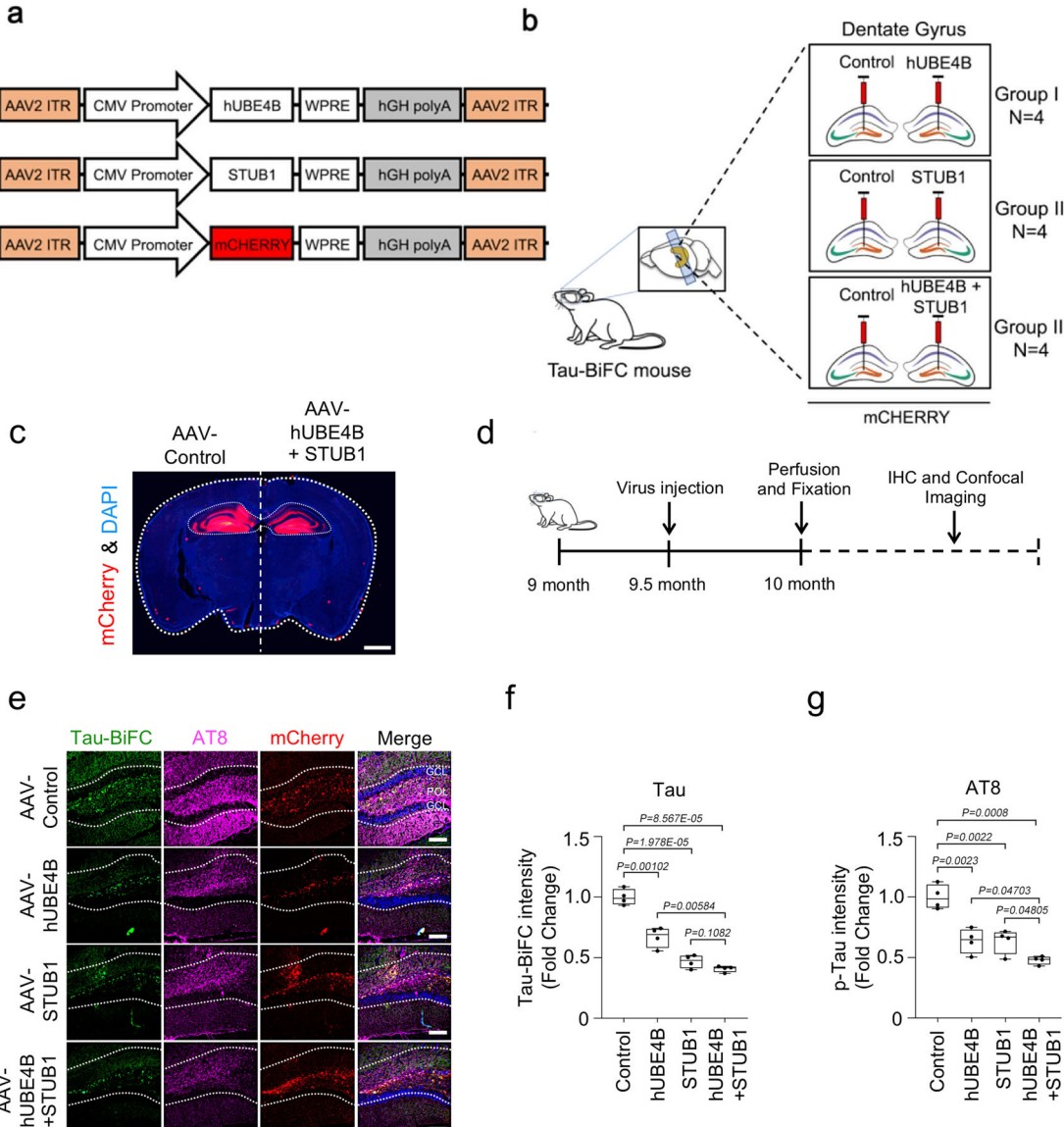

**Fig. 6 Tau oligomers were degraded by UBE4B and STUB1 in the Tau-BiFC mouse model. a** Schematic representation of *AAV-CMV-mCherry* and *AAV-UBE4B-mCherry* virus constructs. **b** Schematic illustration of *AAV-CMV-mCherry* or *AAV-UBE4B-mCherry* virus delivery into the dentate gyrus of Tau-BiFC mice. **c** A fluorescence staining image indicating the foci (red) of *AAV-CMV-mCherry* or *AAV-UBE4B + STUB1-mCherry* virus delivery in the dentate gyrus and hippocampus of Tau-BiFC mice. Scale bar (white), 1 mm. **d** Schematic illustration of the work flow for virus injection and pathological examination in Tau-BiFC mice. **e** *AAV-UBE4B* and *AAV-STUB1* decreased oligomer Tau-BiFC and pTau (S202/T205) levels in the dentate gyrus compared with the control. GCL, granular cell layer; POL, polymorphic layer. Scale bars (white), 80 μm. These experiments were performed four times. **f**, **g** Densitometry analysis revealed that *AAV-UBE4B* and *STUB1* significantly decreased both Tau-BiFC and pTau (S202/T205) levels in the dentate gyrus relative to the control (*AAV-Cont* N = 4; *AAV-UBE4B* N = 4; *AAV-STUB1* N = 4; *AAV-UBE4B + AAV-STUB1*, N = 4; N = 4 biologically independent animals), respectively. In the box plots the whiskers represent the 5th to 95th percentile range. Data are presented as means as ± s.e.m. Statistical significance was determined with two-tailed Student's *t*-test. Statistical source data.

## Methods

**Drosophila culture and stocks**. *Drosophila melanogaster* were maintained at 25 ºC on standard cornmeal, yeast, sugar, and agar medium. *UAS-hTau*, *GMR-GAL4*, and *ElavGAL4* fly lines were obtained from Bloomington Stock Centre (Bloomington, USA). *miR-9a* target RNAi stocks were obtained from Bloomington Stock Centre (Bloomington, USA) and Vienna Drosophila Research Centre (Vienna, Austria). *pUAS-CG11070* and *pUAS-UBE4B* flies were generated by the p-element-mediated germline transformation method with cDNA containing the coding regions of *CG11070* or *UBE4B*.

**Cell culture and transfections**. SH-SY5y neuroblastoma cells were maintained in Dulbecco's modified Eagle's medium (DMEM) supplemented with 10% heat-inactivated fetal bovine serum, penicillin (10 U/mL) and streptomycin (100 μg/mL). Cells were incubated at 37 °C in 5% $CO_2$ and transfected with desired plasmids

using Effectene transfection reagent (Qiagen) following the manufacturer's instructions. SiRNAs were transfected with lipofectamine 2000 transfection reagent.

**Mouse model and virus injection**. Male *TauP301L-BiFC* mice were a kind gift of Dr. Yunkyung Kim (KIST, KOR)[29]. Brain specimens of *TauP301L-BiFC* mice were prepared as previously described[66]. *AAV-CMV-UBE4B* and *AAV-CMV-STUB1* viruses were injected using a stereotaxic micro-injector (Stoelting Co.). Control groups were injected with *AAV-CMV*. Neuropathological experiments were performed at 2 weeks after injection. Mice were housed on a 12:12 h light-dark cycle and maintained at 18–23 °C with humidity between 40 and 60% in pathogen-free facilities at Korea Institute of Science and Technology. All animal experiments were performed in accordance with the National Institutes of Health Guide for the Care and Use of Laboratory Animals of the Korea Institute of Science and Technology.

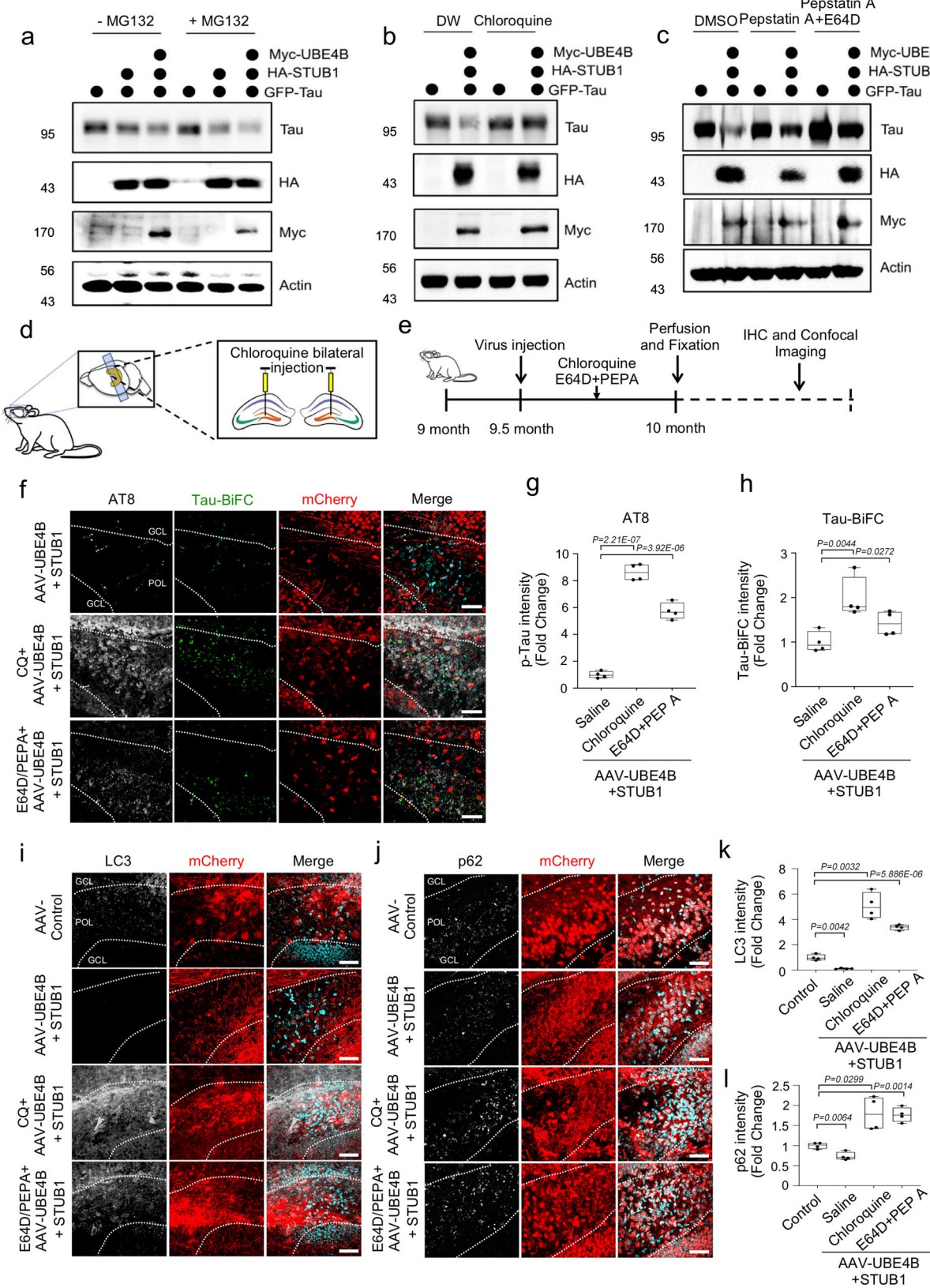

All animal experiments were approved by the Korea Institute of Science and Technology Animal Care Committee.

**Quantification of eye phenotypes in *Drosophila* screens.** *GMR > hTau* flies were crossed with either *UAS-miRNAs*[23] or *miR-9a* target *UAS-RNAi* flies, and the progeny were scored for Tau toxicity in the eyes. Eye images were captured using a

Digiretina 16 camera, and eye size was measured using Image J v1.44 software (National Institutes of Health, Bethesda, USA). Values obtained from these measurements were plotted using a volcano plot in Graphpad Prism 9.1.0.

**Larval crawling assay.** Wandering third instar larvae were briefly washed with PBS to remove residual food. Larvae were dried for a short time on clean filter

**Fig. 7 Tau was degraded by UBE4B and STUB1 primarily via autophagy. a** Treatment of SH-SY5Y cells with the proteasome inhibitor MG132 did not affect Tau degradation mediated by UBE4B/STUB1. **b** Treatment with chloroquine (CQ), an autophagy inhibitor, affected Tau degradation by UBE4B/STUB1. **c** Pepstatin A (PEPA) and E64D, autophagy inhibitors, inhibited Tau degradation by UBE4B/STUB1. All western blots were performed three times. **d** Schematic illustration of autophagy inhibitor injection to the dentate gyrus of Tau-BiFC mice. **e** Schematic illustration of the work flow for autophagy inhibitor injection and pathological examination in Tau-BiFC mice. **f** CQ and E64D plus PEPA (E64D + PEPA) increased pTau (S202/T205) levels in the dentate gyrus relative to saline-injected controls. GCL granular cell layer, POL polymorphic layer. **g**, **h** Densitometry analysis revealed that chloroquine and ED64 + PEPA significantly increased both pTau (S202/T205) and Tau-BiFC levels in the dentate gyrus relative to saline-injected controls (*AAV-UBE4B + AAV-STUB1* (saline control), *N* = 4; *CQ + AAV-UBE4B + AAV-STUB1*, *N* = 4; *E64D/PEPEA + AAV-UBE4B + AAV-STUB1*, *N* = 4; *N* = 4 biologically independent animals). In the box plots the whiskers represent the 5th to 95th percentile range. **i** *AAV-UBE4B* and *AAV-STUB1* decreased LC3 levels in the dentate gyrus relative to control. Scale bars (white): 40 μm. **j** *AAV-UBE4B* and *AAV-STUB1* decreased p62 levels in the dentate gyrus relative to control. Scale bars (white): 40 μm. **k** Densitometry analysis revealed that autophagy inhibitors significantly increased LC3 levels in the dentate gyrus relative to control (*AAV-Control*, *N* = 4; *AAV-UBE4B + AAV-STUB1*, *N* = 4; *CQ + AAV-UBE4B + AAV-STUB1*, *N* = 4; *E64D/PEPA + AAV-UBE4B + AAV-STUB1*, *N* = 4; *N* = 4 biologically independent animals). In the box plots the whiskers represent the 5th to 95th percentile range. **l** Densitometry analysis revealed that autophagy inhibitors significantly increased p62 levels in the dentate gyrus relative to control (*AAV-Control*, *N* = 4; *AAV-UBE4B + AAV-STUB1*, *N* = 4; *CQ + AAV-UBE4B + AAV-STUB1*, *N* = 4; *E64D/PEPA + AAV-UBE4B + AAV-STUB1*, *N* = 4; *N* = 4 biologically independent animals). In the box plots the whiskers represent the 5th to 95th percentile range. Data are presented as means ± s.e.m. Statistical significance was determined with a two-tailed Student's *t*-test. Statistical source data. Each exact *p* value was listed in Statistical source data.

paper and placed on a 2% agar grape juice-coated petri dish. Each genotype was allowed to crawl freely for 90 s. To quantify crawling distance, lines were drawn to track the crawling larvae, and total distance was measured using Image J v1.44 software. Approximately 10–20 animals were tested for each genotype[25,26].

**Immunohistochemistry of neuromuscular junction**. Third instar larvae were dissected in PBS, fixed in 4% formaldehyde in PBS for 15 min, and washed three times in 0.1% Triton X-100 in PBS. FITC-conjugated anti-HRP was used at 1:100 and incubated for 1.5 h at room temperature. Larvae were mounted in Slow Fade Antifade media. Confocal images were captured using Zeiss confocal microscopes. Quantification of the NMJ was performed by counting the number of boutons in each genotype using Image J v1.44 software with cell counter plugin[25,26].

**Quantitative PCR**. The heads of 20 adult *Drosophila* per group were collected and total RNA was isolated with Trizol reagent. After treating the RNA samples with RNase-free DNase I, cDNA was synthesized using the SuperScript III First-Strand Synthesis System (TAKARA, Japan). Quantitative reverse transcription–PCR (qRT–PCR) analysis was performed using a StepOnePlus Sequence Detection System (BioRAD, USA) with SYBR Green PCR Core reagents (BioRAD). Each experiment was performed at least in triplicate. The comparative cycle threshold was utilized to quantify the fold change of each specific mRNA after normalizing to *rp49* levels.

**miRNA–mRNA pull-down assay**. The miRNA–mRNA pull-down assay was performed[23] with minor modifications. Briefly, cells were harvested 24 h after transfection and lysed in lysis buffer (Cell Signaling, USA) containing 20× protease inhibitor (Roche) and 60 U RNaseOUT (Invitrogen). Protein A Dynabeads (Invitrogen, USA) and 2 μg AGO-1-specific antibody were used for immunoprecipitation. The immunoprecipitate was treated with 20 μg/ml proteinase K for 10 min at 37 °C. RNA was extracted using the easy-BLUE kit (iNTRON, Korea), and cDNAs were synthesized with the SuperScript III First-Strand Synthesis System (Invitrogen). To determine if *miR-9a* directly bound *CG11070*, primers that amplified the fragments of its 3′-UTR that included the predicted *miR-9a* seed sequence matches were designed. *Senseless* and *sNPFR1* were used as positive controls and *tubulin* was used as a negative control (Supplementary Table 1).

**Western blot**. Briefly, 20 fly heads for each genotype were homogenized in RIPA buffer, and lysates were loaded in each lane of 10% SDS gels and transferred to nitrocellulose membrane. Membranes were blocked in 5% BSA and incubated with primary antibodies at 4 °C overnight. After washing membranes with TBS-T, membranes were incubated with the appropriate secondary antibody. Using the ECL Western blotting detection reagent, membranes were developed and images were captured using FluorChem E image processor. Antibodies used were anti-Tau (1:1000, T46, Cat no. 13-6400, Invitrogen), anti-AT180 (1:1000, Cat no. MN1040, Invitrogen), anti-PHF-1 (1:1000, Cat no. MN1050, Invitrogen), anti-AT8 (1:1000, Cat no. MN1020, Invitrogen) and anti-β-actin (1:1000, Cat no. JLA20, DHSB). β-actin was used as a loading control. Signal intensity was quantified using ImageJ (NIH) software. Flies used were 30 days old after eclosion. For Western blot analysis of SH-SY5y cells, the following antibodies were used: anti-Tau (Cat no. ab64193, Abcam), anti-β-actin (Cat no. LF-PA0207, AB Frontier), anti-Myc (Cat no. C3956, Sigma), anti-HA (Cat no. H6908, Sigma), and anti-CHIP (Cat no. sc-133066, Santa Cruz Biotechnology).

**Immunoprecipitation**. Cells were transfected with the indicated plasmids. After 24 h of transfection, cells were harvested and lysed with IP buffer (50 mM HEPES

pH 7.5, 150 mM NaCl, 1.5 mM MgCl$_2$, 5 mM KCl, 0.1% Tween-20, 2 mM DTT, and protease inhibitor cocktail (Roche)). Lysates were centrifuged at 9700 × *g* for 30 min at 4 °C, and the collected supernatants were incubated with anti-HA agarose beads (Sigma) at 4 °C for 4 h. The beads were then washed with buffer containing 50 mM HEPES pH 7.5, 150 mM NaCl, 1.5 mM MgCl$_2$, 5 mM KCl, 0.1% Tween-20, and 2 mM DTT, and the bound proteins were eluted with 2× SDS sample buffer. Samples were quantified by Western blot after heating at 95 °C for 10 min.

**His-ubiquitin pull-down assay**. PCS2-His-ubiquitin was co-transfected with the indicated plasmids and after 24 h of transfection, transfected cells were treated with 10 μM MG132 for an additional 6 h. Cells were lysed in urea lysis buffer (8 M Urea, 0.3 M NaCl, 0.5 M Na$_2$HPO$_4$, 0.05 M Tris, 0.001 M PMSF, 0.01 M imidazole, pH 8.0) and sonicated for 4 min. Cell lysates were transferred to equilibrated Ni-NTA agarose and incubated for 4 h at room temperature. Beads were then washed five times with urea wash buffer (8 M Urea, 0.3 M NaCl, 0.5 M Na$_2$HPO$_4$, 0.05 M Tris, 0.001 M PMSF, 0.02 M imidazole, pH 6.5) and conjugated proteins were eluted with 40 μL 2X Laemmli/Imidazole (200 mM imidazole). Eluted proteins were analyzed by Western blotting after heating the samples at 95 °C for 10 min.

**Protein stability analysis**. SH-SY5y cells were transfected with the indicated plasmids or siRNAs, and treated with 100 μg/mL cyclohexamide (CHX) or vehicle after 24 h of transfection. Cells were collected at the specified time points after CHX treatment and immunoblotted with antibodies against the specified proteins. To evaluate autophagic degradation, *Tau* was co-transfected with *STUB1* and *UBE4B* for 24 h, and cells were treated with 50 μM Chloroquine or vehicle for 8 h. Cells were collected and lysed, and lysates were immunoblotted with antibodies against the specified proteins.

**In vivo confocal microscopy and image analyses**. Immunofluorescence staining for anti-Tau 5 (1:200, ab3931, Abcam), anti-pTau AT8 (1:200, Cat no. MN1020, Invitrogen), anti-pTau AT180 (1:200, Cat no. MN1040, Invitrogen), anti-pTau PHF-1 (1:200, Cat no. MN1050, Invitrogen), anti-LC3 (1:200, Cat no. M152-3, MBL), anti-P62 (1:200, Cat no. PM045, MBL) and anti-Beclin (1:200, Cat no. PD017, MBL) was performed in Tau-BiFC mice models[29]. Fluorescence was observed by confocal microscopy (Nikon A1R, JAPAN). Pre-absorption with excess target protein or omission of primary antibody was used to demonstrate antibody specificity and remove background generated by the detection assay. Co-localization and quantitative assessment of images were conducted using NIH Image J v1.44 software.

**Statistics and reproducibility**. All experiments were performed more than three times. In case of animal study, 'N' represents the number of biologically independent animals. Boxplots were generated using the standard style except that the whiskers represent minimum to maximum. In bar charts, unless otherwise noted, data are presented as mean ± SEM, and comparisons between groups were conducted using the Student's *t*-test considering *p* < 0.05 to be statistically significant. For multiple comparisons, we performed a one-way analysis of variance followed by pairwise *t*-tests using the Bonferroni method to adjust the *p*-value threshold for significance.

**Reporting summary**. Further information on research design is available in the Nature Research Reporting Summary linked to this article.

## Data availability
The data that support the findings of this study are available from the corresponding author upon reasonable request. Source data are provided with this paper.

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

## Acknowledgements

*Drosophila* stocks were obtained from Bloomington Sock Center (Bloomington, IN, USA) and Vienna *Drosophila RNAi* Center (VDRC, Vienna, Austria). This work was supported by grants from KRIBB Research Initiative Program, National Research Council of Science & Technology (CRC-15-04-KIST), KIST (2E30951, 2E30954 and 2E30962), and National Research Foundation of Korea (2015R1A5A1009024, 2017HID3A1A02054608, 2018M3C7A1056894, 2019R1A2C 2004052, 2019R1A2C2004149, 2020R1A4A4079494, and 2020M3E5D9079742).

## Author contributions

M.S., E.J.S., H.R., and K.Y designed the research, M.S., S.J.H., T.D., and Y.S.S. performed experiments, M.S., Y.K.K., J.L., E.J.S., H.R., and K.Y. analyzed the data, and M.S., E.J.S., H.R., and K.Y. wrote the manuscript.

## Competing interests

The authors declare no competing interests.
