## [Peer Review File · Nature Communications]

REVIEWER COMMENTS

Reviewer #1 (Remarks to the Author):

This is a beautiful paper that shows an effect of tau degradation. It is very well written and the data are clear. However, certain controls are missing and additional autophagy markers have to be shown to conclude that autophagy is the main player here, especially that previous evidence suggest that both autophagy and the proteasome are means of tau degradation.

Some clarification is needed:

Does miRNA-9 target any other genes- ?

Is there a known pathway that is affected by miRNA-9

What other miRNAs are known to control the UBE genes?

If both total and p-tau are reduced, what are the consequences on the homeostasis of neurons that require tau for microtubules?

It appears from the data that ubiquitination does not depend on hyper-phosphorylation, i.e. total tau does not have to be post-translationally modified to be ubiquitinated and degraded. This is a conundrum. Please discuss- we need total intact tau and bad ptau should be degraded and that may depend on ubiquitination.

Does miRNA-9 control STUB expression? Why did you choose to combine STUB with UBE4?

Please discuss the difference between E3 and E4 ubiquitin ligases and why this is important in terms of tau ubiquitination.

Results

Lane 184-189- Due to the essential regulatory roles of these ubiquitin ligases in protein degradation, we examined UBE4B and STUB1-mediated Tau degradation. By inhibiting protein synthesis with cycloheximide (CHX), we observed that Tau degradation was increased by STUB1 (Fig. 4b, lanes f-I and Fig. 4c) and UBE4B overexpression (Fig. 4b, lanes 1-4 and Fig. 4c), and was further increased by co-expression of STUB1 and UBE4B (Fig. 4b, lanes 6-9 relative to lanes 1-4 and Fig. 4c). However, if you inhibit protein synthesis tau synthesis is also reduced so what is the control here? There are no control experiments and therefore, an assumption that UBE and STUB1 increased tau degradation is not feasible.

Additionally, there is a contradiction with the data when UBE and STUB are co-expressed tau is reduced, but STUB1 RNAi silencing does not affect tau degradation. The assumption that UBE enhances STUB activity is not supported by data.

Lane 228- MG132 or the ALS inhibitor Chloroquine. Interestingly, MG132 treatment did not inhibit Tau degradation relative to the control cells (Fig. 6a). Please discuss in the light of previous data showing that Tau is reduced via the proteasome?

Contrastingly, Chloroquine significantly inhibited Tau degradation (Fig. 6b), suggesting that ALS was the major pathway of UBE4B and STUB1-mediated Tau degradation in 231 neuroblastoma cells. Inhibition of autophagy via chloroquine has to be demonstrated via certain biomarkers like LC3, beclin, not only P62.. not all autophagy is dependent on P62 accumulation and P62 is more a sensor of ubiquitination of proteins, therefore it is difficult to conclude that changes in ubiquitination does not affect the proteasome but the ALS. More markers of autophagy have to be demonstrated.

Discussion

Please discuss previous evidence that tau is degraded via both autophagy and the proteasome.

Figure 1, 2, 3, 4- what is the N? were these experiments replicated.

Reviewer #2 (Remarks to the Author):

This manuscript by Subramanian et al uses the Drosophila system to screen for miRNAs that, when overexpressed, suppress the effects of ectopic hTau. From this, they connect miR-9 and its predicted

target CG11070 to Tau phenotypes in flies. Then, extending this work into the human cell culture and mouse systems, show that overexpression of mammalian orthologs of CG11070 can promote the degradation of Tau. Overall, the text is clearly written, the experiments are well presented and, for the most part, appropriately interpreted, and the conclusions are well founded. Addressing the comments below, including importantly experiments that analyze the effects of miR-9 loss on hTau phenotype, would strengthen the manuscript.

1. The authors should refer to effects that miRNAs have on eye size in the absence of hTau to strengthen the conclusion on lines 111-112 implicating miR-9 family in Tau toxicity. As currently written, it is possible to infer that miRNA may have an equivalent effect on eye size in the absence of Tau. Figs 1C, 1D and S1 should be better described.
2. The meaning of the greyscale colorcoding in Fig S1 and S2 is not explained in the legends.
3. A reference for the mRNA-miRNA pulldown assay mentioned on line 132 should be included.
4. I am not convinced that the mRNA-miRNA pulldown shows direct interaction between miR-9 and CG11070. It certainly is supportive evidence, but is it possible that other miRNAs recruit CG11070 to Ago1 complex. I think the strongest conclusion is that miR-9 and CG11070 co-precipitate with Ago1.
5. Does knockdown of the Drosophila Stub1 ortholog eliminate the UAS-CG11070 suppression of hTau phenotypes?
6. Does reduction of miR-9 (with mutant or sponge) suppress hTau phenotypes in the same way the UAS-CG11070 does? This is an important experiment to support the authors general model about the relationship between miR-9- CG11070 and hTau
7. The authors suggest on line 252 that they knocked down miR-9a but I failed to find this experiment in the results.

Reviewer #3 (Remarks to the Author):

In this manuscript, Subramanian et. al. claim that the miRNA9-UBE4b-autophagy pathway regulates Tau toxicity. After identifying that miRNA-9 modifies hTau in a drosophila human Tau overexpression model by genome-wide miRNA screening, the authors further demonstrated that its target gene CG11070, which is an orthologue to UBE4B in mammals, promotes Tau degradation through autophagy-mediated pathways in drosophila and mice. This is an interesting finding. There are, however, there are several concerns and clarifications that are needed regarding this work.

Major

1. The Drosophila miRNA library (Fig 1D & Supp.1) shows that the eye sizes are already reduced in UAS-miR-9a,b,c lines (compared to GMR-GAL4) without the human Tau overexpression. This seems to indicate that miR-9 family per se is also involved in regulating eye size regardless of human Tau dependent neurodegeneration, through other pathways, such as development pathways or hTau-independent neurodegeneration. This issue at least needs to be discussed.
2. The central finding in this study is that UBE4B promotes Tau degradation through an autophagy-mediated pathway. However, this conclusion is not convincing with the current data set.
 - a. Authors used Tau-BiFC animal model to visualize/monitor Tau aggregation in mice. This mouse model is designed to track Tau oligomerization, and other markers used detect both tau monomer and oligomer. Concerns of interpretation arise in that UBE4B/STUB1 still may mediate monomeric tau degradation through the ubiquitin proteasome system, but this may not be reflected in the current assays that are specifically assessing oligomeric tau changes.

- b. In figure 6, authors ran an in-vivo chloroquine injection experiment to prove that UBE4B/STUB1-promoting degradation mediated autophagy. Authors need to inject chloroquine in the Tau-BiFC::UBE4B/STUB1 overexpressing mouse, not just Tau-BiFC mice to see if chloroquine is rescuing the Tau degradation facilitated by UBE4B/STUB1 overexpression.
- c. Also, chloroquine administration is not sufficient to prove the autophagy mediation. It is true that chloroquine is known to inhibit autophagic activity, but its mechanism in vivo is not fully understood and other mechanisms, such as inflammatory modulation, are involved, which may also modulate tau pathology. The interpretation of this experiment should at least be softened.
- d. In addition, authors should measure a direct autophagosome formation marker (e.g. LC3), rather than only p62 which may cross into the UBS pathway.
3. In general, the images provided in the manuscripts (in particular, in vivo studies) are not adequate and difficult to evaluate. For example, in the figure 5, it seems that Tau-BiFC is mostly expressed in the interneurons in the hilus of the DG. On the other hand, AT8 staining shows expression throughout the DG. This is a concern with the high concentration of primary antibody application (1:200). The authors need to show more controls for the AT8 staining.
- a. In addition, in the figures, the fluorescence intensities are different between supra- and infra-pyramidal blades in different groups (even in DAPI). This might be due to the variations in injection sites and area affected by virus transfections. The authors need to provide injection sites and spreading area including control groups.
- b. Related to the issue above, authors need to provide the detailed experiment scheme with age and time line, to know what stage of tau pathology has been affected.
4. In figure 5 and 6, it was not clearly described how the number of subjects (n=4) and cell counting subjects (n=30 for figure 5 and n=20 for figure 6) were generated. The cell counting/intensity of staining needs to be quantified and averaged as an individual subject level (n=4 per group) and treat as a sample for statistical analysis (instead of using multiple sections from an individual subject as separate n's).
5. The authors claim that the UBE4B and STUB1, but not alone, co-regulate tau degradation, in evidence by figure 4 (Tau was significantly increased by co-expression of UBE4B and STUB1, but not by expression of UBE4B or STUB1 alone). This argument is not consistent, necessarily, in the in-vivo study (figure 5). Either UBE4B or STUB1 over-expression alone also reduced the tau levels. Can you authors explain this?

Minor

1. "Mohanty et al. (unpublished) demonstrated, line 320, that UBE4B ligase is also capable of both Lys48 and Lys63 polyubiquitination.", please discuss this more based on published data.
2. "The most significant reductions in eye sizes were induced by overexpression of miR-9a, miR-9b, and miR-9c". miRNA- 989 and 932 also reduced eye size even more than miR-9a, miR-9b.
3. Figure 4c. please report statistics using a repeated measure.

RESPONSE TO REVIEWERS' COMMENTS

Reviewer #1 (Remarks to the Author):

This is a beautiful paper that shows an effect of tau degradation. It is very well written and the data are clear. However, certain controls are missing and additional autophagy markers have to be shown to conclude that autophagy is the main player here, especially that previous evidence suggest that both autophagy and the proteasome are means of tau degradation.

Some clarification is needed:

Does miRNA-9 target any other genes- ?

Response: *miRNA-9* has other targets apart from UBE4B/CG11070. We used three different microRNA target prediction programs and narrowed down the *miR-9* targets to 34 mRNAs (Supplementary Fig.2a) and tested these target RNAi flies in the eye size phenotypes (Supplementary Fig.2b, c). Among 34 *miR-9* targets, *CG11070/UBE4B-RNAi* flies showed the strongest genetic interaction with the *GMR>hTau^{WT}* flies (Figure 2a-d, Supplementary Fig.2c).

Is there a known pathway that is affected by miRNA-9

Response: *miRNA-9* is involved in the regulation of body size through sNPFR pathway in *Drosophila*. Apart from this, *miRNA-9* is also involved in the process of neurogenesis in mammals through the senseless gene. We have addressed this in the discussion section, page 15-16 of the revised manuscript as seen below. We have identified a new mechanism of *miR-9* in regulating Tau toxicity through the autophagy dependent pathway.

“Functional analysis of *miR-9* has revealed its involvement in the regulation of neuronal progenitor cells and differentiation of neuronal cells during development. In AD patients, *miR-9* expression is elevated. In *Drosophila*, *miR-9a* plays an important role in the development of sensory organs by suppressing its target gene *senseless* and also regulates body growth through sNPFR signaling.”

What other miRNAs are known to control the UBE genes?

Response: In addition to *miR-9*, *UBE4B* is also regulated by *miR-26*, *miR-148/miR-152* and *miR-15/16/195/424/497*. We added this information in page 8 of the revised manuscript.

If both total and p-tau are reduced, what the consequences are on the homeostasis of neurons that require tau for microtubules?

Response: We agree with the reviewer's comment regarding tau depletion and microtubule stability. In addition to total Tau, MAP6 protein also stabilizes the microtubules. Recent reports have shown that the reduction of Tau proteins makes the labile domain of microtubule more stable. In our studies, we also saw similar phenotypes, which are a reduction of total tau rescued tauopathy phenotypes in the *Drosophila* model system. We have added this information in the discussion section of page 16-17 of the revised manuscript as seen below.

“Since tau is involved in maintaining the stability of microtubules in neurons, the degradation of tau proteins affects microtubule stability as seen in tauopathy and Alzheimer's disease models. However, recent studies have shown that reduction in Tau proteins leads to increased accumulation of MAP6 proteins on the microtubules and enhances the stability of microtubules in neurons. The rescue phenotype seen by *UBE4B/CG1170* may be a result of compromised function of Tau and MAP6 proteins in the neurons.”

It appears from the data that ubiquitination does not depend on hyperphosphorylation, i.e. total tau does not have to be post-translationally modified to be ubiquitinated and degraded. This is a conundrum. Please discuss- we need total intact tau and bad ptau should be degraded and that may depend on ubiquitination.

Response: From our data, we show that ubiquitination is essential for the clearance of tau proteins and reduces phosphorylated tau levels which we have shown in Figure.4a. Since ubiquitination is the key component in protein modification for degradation, ubiquitination alone acts as a triggering factor for degradation and phosphorylation of Tau proteins affect ubiquitination and eventually degradation. Overexpression of both *UBE4B* and *STUB1* gene increases ubiquitination of Tau proteins and reduces phosphorylation of Tau proteins. The reduction of pTau levels may be due to reduced levels of total Tau proteins. Studies have shown that the acetylation of Tau increases with an elevated p-Tau level and affects Tau degradation in tauopathies. Both ubiquitination and acetylation shares common Lys residue Lys280 which is involved in stability of Tau proteins. We add this

information in the discussion section of page 17 of the revised manuscript as seen below.

“Reduction in Tau proteins leads to reduced pTau levels in *UBE4B/CG11070* rescue animals, due to increased ubiquitination of Tau proteins. Studies have shown that the acetylation of Tau inhibits degradation of phosphorylated Tau and contributes to increased tauopathy. Both ubiquitination and acetylation shares common Lys residue Lys280⁴⁶ which is involved in stability of Tau proteins.”

Does miRNA -9 control STUB expression? Why did you choose to combine STUB with UBE4? Please discuss the difference between E3 and E4 ubiquitin ligases and why this is important in terms of tau ubiquitination.

Response: *miRNA-9* does not control STUB expression. The reason we chose STUB1 was that STUB1 was known to be involved in ubiquitination of Tau proteins and clearance. Since UBE4B is involved in ubiquitination, which is a novel protein, we wanted to study this effect with the known E3 ligases, which will enhance the ubiquitination of Tau proteins and degradation. Hence, we used STUB1 function in the background of UBE4B on Tau ubiquitination. We added this information in the results section of page 10 of the revised manuscript as seen below.

“Previous studies have demonstrated that UBE4B has E4 ubiquitin ligase activity, and that Tau is ubiquitinated by STUB1 E3 ligase. STUB1 was chosen due to its role in ubiquitination of Tau proteins and is not regulated by miR-9.”

Results

Lane 184-189- Due to the essential regulatory roles of these ubiquitin ligases in protein degradation, we examined UBE4B and STUB1-mediated Tau degradation. By inhibiting protein synthesis with cycloheximide (CHX), we observed that Tau degradation was increased by STUB1 (Fig. 4b, lanes f-l and Fig. 4c) and UBE4B overexpression (Fig. 4b, lanes 1-4 and Fig. 4c), and was further increased by co-expression of STUB1 and UBE4B (Fig. 4b, lanes 6-9 relative to lanes 1-4 and Fig. 4c). However, if you inhibit protein synthesis tau synthesis is also reduced so what is the control here? There are no control experiments and therefore, an assumption that UBE and STUB1 increased tau degradation is not feasible.

Response: We appreciate the reviewer’s comments. Cycloheximide (CHX) experiment is widely used for determining the half-life of the protein because it provides the degradation pattern of the protein without the effect of transcription or

translation. In this study, we found that Tau is degraded by STUB1 and UBE4B. Therefore, Tau levels are already low in the co-overexpression of STUB1 and UBE4B, even at 0 hr of CHX treatment (Fig.4a, lane6).

To clearly show that STUB1 and UBE4B increased Tau degradation, we tested Tau stability following Chloroquine (CQ) treatment to prevent STUB1 and UBE4B mediated prompt lysosomal Tau degradation before the CHX effect. As shown in the supporting figure (below), Tau protein degradation was gradually increased by STUB1 and UBE4B over 4h of cycloheximide treatment while no early Tau degradation was observed at 0h point, which provides the control for our assumption that STUB1 and UBE4B increased Tau degradation over the cycloheximide treatment.

Additionally, there is a contradiction with the data when UBE and STUB are co-expressed tau is reduced, but STUB1 RNAi silencing does not affect tau degradation. The assumption that UBE enhances STUB activity is not supported by data.

Response: Tau degradation was significantly increased by the overexpression of UBE4B alone (Fig 4b lanes 1-4 relative to lanes a-d) or co-expression with STUB1 (Fig 4b lanes 6-9 relative to lanes a-d) in SH-SY5Y cells. We also found that STUB1 activity is important for UBE4B-mediated Tau ubiquitination as co-expression with a dominant-negative mutant (STUB1^{H260Q}) failed to enhance Tau ubiquitination (Fig. 4a, lane 6). Therefore, we depleted STUB1 to eliminate the effect of any STUB1 on UBE4B mediated Tau degradation and found that UBE4B overexpression no longer affects Tau degradation (Supplementary Figure 5b). We added the quantitation data in Supplementary Figure. 5c. These results suggest that UBE4B does not directly affect Tau degradation but acts as a critical factor to facilitates STUB1 activity for the ubiquitination and degradation of Tau. We add this

information in the results section of page 11 of the revised manuscript as seen below.

“Because *UBE4B* overexpression alone degraded Tau (Fig. 4b,c), we knocked down endogenous *STUB1* with siRNA (Supplementary Fig. 5a) and examined UBE4B-mediated Tau degradation (Supplementary Fig. 5b-c). Interestingly, *UBE4B* overexpression did not affect Tau degradation when *STUB1* was knocked down (Supplementary Fig. 5b, lanes 6–9 compared with lanes 1–4 and Supplementary Fig. 5c).”

Lane 228- MG132 or the ALS inhibitor Chloroquine. Interestingly, MG132 treatment did not inhibit Tau degradation relative to the control cells (Fig. 6a (revised Fig.7a)). Please discuss in the light of previous data showing that Tau is reduced via the proteasome?

Response: We have added this information in the discussion section of page 17-18, of the revised manuscript as seen below.

“Studies on Tau degradation have shown that the proteasomal pathway plays a crucial role. Inhibition of proteasome in HEK cells increases the levels of full length Tau proteins. Similar results were seen in SH-SY5Y cells with increased accumulation of both full length Tau protein and mutant P301L Tau protein levels when proteasome degradation is inhibited. However, from our studies, *STUB1*- and *UBE4B*-mediated Tau degradation was inhibited by the autophagy inhibitors such as Chloroquine, PEPA, and E64D (Fig. 7b and c) but not by the proteasomal inhibitor MG132 (Fig. 7a), suggesting that *STUB1/UBE4B*-mediated Tau degradation was facilitated by autophagy rather than the UPS.”

Contrastingly, Chloroquine significantly inhibited Tau degradation (Fig. 6b (revised Fig.7b)), suggesting that ALS was the major pathway of *UBE4B* and *STUB1*-mediated Tau degradation in 231 neuroblastoma cells. Inhibition of autophagy via chloroquine has to be demonstrated via certain biomarkers like LC3, beclin, not only P62.. not all autophagy is dependent on P62 accumulation and P62 is more a sensor of ubiquitination of proteins, therefore it is difficult to conclude that changes in ubiquitination does not affect the proteasome but the ALS. More markers of autophagy have to be demonstrated.

Response: As suggested by reviewer 1, we performed immunofluorescence staining with more autophagy markers such as LC3 and Beclin. New data was added in Fig. 7i-k and Supplementary Fig. 8a and b of the revised manuscript. New

results were added to the result section of page 14 of the revised manuscript as seen below.

“Because autophagy inhibitor treatment blocked autophagic Tau degradation *in vitro*, we tested this in the *in vivo* system by injecting autophagy inhibitors into the dentate gyri of *Tau-BiFC* mice with co-expression of *UBE4B* and *STUB1* (Fig. 7d and e). We measured oligomeric Tau and phosphorylated Tau levels in the dentate gyrus of *Tau-BiFC* mice with the AT8 antibody (S202/T205) (Fig. 7f-h). Inhibition of ALS by Chloroquine and E64D+PEPA significantly increased oligomeric and phosphorylated Tau (S202/T205) (Fig. 7f-h). Similarly, the phosphorylated forms of Tau p-S396/S404 (PHF-1) and p-T231 (AT180) were also significantly increased in Chloroquine and E64D+PEPA-treated *Tau-BiFC* mice (Supplementary Fig. 7a-d). Since ALS disruption changes LC3 and p62 levels, which are inversely correlated with the efficiency of autophagy, we measured LC3 and p62 levels in our model system. Co-expression of *UBE4B* and *STUB1* significantly decreased LC3 and p62 levels relative to the control in the dentate gyri of *Tau-BiFC* mice, whereas autophagy inhibitors elevated LC3 and p62 levels relative to the control in the dentate gyri of *UBE4B+STUB1*-expressed *Tau-BiFC* mice (Fig. 7i-l). Furthermore, autophagy inhibitors modulated BECN1 levels compared to the control in the dentate gyri of *UBE4B+STUB1*-expressed *Tau-BiFC* mice (Supple Fig. 8). Collectively, these results suggested that monomeric and oligomeric Tau degradations by *STUB1* and *UBE4B* are mediated through the autophagy pathway rather than the ubiquitin proteasome system.”

Discussion

Please discuss previous evidence that tau is degraded via both autophagy and the proteasome.

Response: We have added this information in the discussion section of page 17-18 of the revised manuscript as seen below.

“Studies on Tau degradation have shown that the proteasomal pathway plays a crucial role. Inhibition of proteasome in HEK cells increases the levels of full length Tau proteins. Similar results were seen in SH-SY5Y cells with increased accumulation of both full length Tau protein and mutant P301L Tau protein levels when proteasome degradation is inhibited. However, from our studies, *STUB1*- and *UBE4B*-mediated Tau degradation was inhibited by the autophagy inhibitors such as Chloroquine, PEPA, and E64D (Fig. 7b and c) but not by the proteasomal inhibitor MG132 (Fig. 7a), suggesting that *STUB1/UBE4B*-mediated Tau degradation was facilitated by autophagy rather than the UPS.”

Figure 1, 2 3, 4- what is the N? were these experiments replicated.

Response: We have added the N in the figure legends of the revised manuscript.

Reviewer #2 (Remarks to the Author):

This manuscript by Subramanian et al uses the *Drosophila* system to screen for miRNAs that, when overexpressed, suppress the effects of ectopic hTau. From this, they connect miR-9 and its predicted target CG11070 to Tau phenotypes in flies. Then, extending this work into the human cell culture and mouse systems, show that overexpression of mammalian orthologs of CG11070 can promote the degradation of Tau. Overall, the text is clearly written, the experiments are well presented and, for the most part, appropriately interpreted, and the conclusions are well founded. Addressing the comments below, including importantly experiments that analyze the effects of miR-9 loss on hTau phenotype, would strengthen the manuscript.

1. The authors should refer to effects that miRNAs have on eye size in the absence of hTau to strengthen the conclusion on lines 111-112 implicating miR-9 family in Tau toxicity. As currently written, it is possible to infer that miRNA may have an equivalent effect on eye size in the absence of Tau. Figs 1C, 1D and S1 should be better described.

Response: Over-expression of miRNAs in the absence of hTau expression alone shows significant reduction in eye size as these miRNAs may have targets, which regulate the eye size in *Drosophila*. However, when these miRNAs were expressed in the presence of hTau, the reduction of eye size was enhanced drastically. We have added this information in the result section of page 6 of the revised manuscript as seen below.

“The reduced eye size of *miR-9a*, *miR-9b* or *miR-9c* in the absence of *hTau* expression (Fig. 1c, d, and Supplementary Fig. 1) may be due to the involvement of these *miRNAs* during eye development where these miRNAs may regulate other target genes and affect morphology and development of eye sizes. However, when these *miRNAs* were expressed in the presence of *hTau*, the reduction of eye size was enhanced drastically.”

2. The meaning of the greyscale colorcoding in Fig S1 and S2 is not explained in the legends.

Response: The greyscale colorcoding does not mean anything and we changed a single tone of greyscale in the revised manuscript.

3. A reference for the mRNA-miRNA pulldown assay mentioned on line 132 should be added.

Response: We have mentioned the reference in the results section of page 7, line 20 of the revised manuscript.

4. I am not convinced that the mRNA-miRNA pulldown shows direct interaction between miR-9 and CG11070. It certainly is supportive evidence, but is it possible that other miRNAs recruit CG11070 to Ago1 complex. I think the strongest conclusion is that miR-9 and CG11070 co-precipitate with Ago1.

Response: We used scrambled miRNA for control. If other miRNAs recruit CG11070 to Ago1 complex, the enrichment of target genes was not changed when compared with miR-9a transfection. The negative control tubulin showed similar levels of enrichment in scrambled and miR-9a samples. It means our pull-down assay was working well. We have added this data in Fig.2e and in the results section of page 7 of the revised manuscript as seen below.

“Similar to the known *miR-9a/miR-9* targets *senseless*²⁴ and *sNPFR1*²³, we found that *miR-9a* bound to and enriched *CG11070* mRNA in *Drosophila* S2 cells, compared to its control scrambled miRNA (Fig. 2e).”

5. Does knockdown of the *Drosophila* *Stub1* ortholog eliminate the UAS-CG11070 suppression of hTau phenotypes?

Response: We agree with the reviewer’s comment about the importance of *STUB1* knockdown in the *in vivo* model system. When we knockdown *STUB1* by UAS-*STUB1-RNAi* in *UAS-hUBE4B+UAS-hTau* background, alleviated phenotypes of eye size, larval crawling, and NMJ bouton numbers by *UAS-hUBE4B+UAS-hTau* compared with those of *UAS-hTau* were abolished. We have added this data in the Fig. 5 and in the results section of page 12 of the revised manuscript as seen below.

“Studies from *in vivo* model system showed that the knockdown of *STUB1* gene in flies expressing *GMR>hTau+hUBE4B* in the eyes, significantly reduced eye

phenotype when compared with *GMR>hTau+hUBE4B* flies (Fig.5a & b). Similarly the knockdown of *STUB1* in neurons expressing *Elav>hTau+hUBE4B* also significantly reduced larval locomotion phenotype when compared with *Elav>hTau+hUBE4B* in neurons (Fig. 5c & d). However, NMJ phenotype showed no change in larvae expressing *Elav>hTau+hUBE4B+STUB1-RNAi* when compared with *Elav>hTau+hUBE4B* larvae (Fig. 5e & f).”

6. Does reduction of miR-9 (with mutant or sponge) suppress hTau phenotypes in the same way the UAS-CG11070 does? This is an important experiment to support the authors general model about the relationship between miR-9- CG11070 and hTau

Response: We agree with the reviewer’s comment about *miR-9a* mutant or sponge suppresses *UAS-CG11070* phenotype. *miR-9a-sponge* significantly suppressed *hTau* larval phenotypes of locomotion and NMJ bouton numbers similar to *UAS-CG11070* and *UAS-hUBE4B* overexpression in neurons. We have added this data in the Supplementary Fig.4 and in the results section of page 9 of the revised manuscript as seen below.

“The knockdown of *miR-9a* by *miR-9a-sponge* (SP) showed no change in the eye sizes when compared to *GMR>hTau* flies (Supplementary Fig. 4a & b). However, knockdown of *miR-9a-SP* in neurons of *Elav>hTau* rescued the larval locomotion phenotype similar to the overexpression of *CG11070* and *hUBE4B* (Supplementary Fig. 4c & d). Similarly, the knockdown of *miR-9a-SP* in neurons also rescued NMJ bouton numbers when compared with the overexpression of *CG11070* and *hUBE4B* in neurons (Supplementary Fig. 4e & f). This data also confirmed that *miR-9a* and *CG11070/UBE4B* forms a common axis involved in regulating Tau toxicity in *Drosophila*.”

7. The authors suggest on line 252 that they knocked down miR-9a but I failed to find this experiment in the results.

Response: We have added this data in the Supplementary Fig.4 and in the results section of page 9 of the revised manuscript.as seen below.

“The knockdown of *miR-9a* by *miR-9a-sponge* (SP) showed no change in the eye sizes when compared with *GMR>hTau* flies (Supplementary Fig. 4a & b). However, the knockdown of *miR-9a-SP* in neurons of *Elav>hTau* rescued the larval locomotion phenotype similar to the overexpression of *CG11070* and *hUBE4B* (Supplementary Fig. 4c & d). Similarly, the knockdown of *miR-9a-SP* in neurons

also rescued NMJ bouton numbers when compared to the overexpression of *CG11070* and *hUBE4B* in neurons (Supplementary Fig. 4e & f). This data also confirmed that *miR-9a* and *CG11070/UBE4B* forms a common axis involved in regulating Tau toxicity in *Drosophila*.”

Reviewer #3 (Remarks to the Author):

In this manuscript, Subramanian et. al. claim that the miRNA9-UBE4b-autophagy pathway regulates Tau toxicity. After identifying that miRNA-9 modifies hTau in a drosophila human Tau overexpression model by genome-wide miRNA screening, the authors further demonstrated that its target gene *CG11070*, which is an orthologue to *UBE4B* in mammals, promotes Tau degradation through autophagy-mediated pathways in drosophila and mice. This is an interesting finding. There are, however, there are several concerns and clarifications that are needed regarding this work.

Major

1. The *Drosophila* miRNA library (Fig. 1D & Supp.1) shows that the eye sizes are already reduced in UAS-miR-9a,b,c lines (compared to GMR-GAL4) without the human Tau overexpression. This seems to indicate that miR-9 family per se is also involved in regulating eye size regardless of human Tau dependent neurodegeneration, through other pathways, such as development pathways or hTau-independent neurodegeneration. This issue at least needs to be discussed.

Response: Over-expression of miRNAs in the absence of hTau expression alone shows significant reduction in eye size as these miRNAs may have targets, which regulate the eye size in *Drosophila*. However, when these miRNAs were expressed in the presence of hTau, the reduction of eye size was enhanced drastically. We have added this information in the result section of page 6 of the revised manuscript as seen below.

“The reduced eye size of *miR-9a*, *miR-9b* or *miR-9c* in the absence of *hTau* expression (Fig. 1c, d, and Supplementary Fig. 1) may be due to the involvement of these *miRNAs* during eye development where these miRNAs may regulate other target genes and affect morphology and development of eye sizes. However, when these *miRNAs* were expressed in the presence of *hTau*, the reduction of eye size was enhanced drastically.”

2. The central finding in this study is that *UBE4B* promotes Tau degradation through an

autophagy-mediated pathway. However, this conclusion is not convincing with the current data set.

a. Authors used Tau-BiFC animal model to visualize/monitor Tau aggregation in mice. This mouse model is designed to track Tau oligomerization, and other markers used detect both tau monomer and oligomer. Concerns of interpretation arise in that UBE4B/STUB1 still may mediate monomeric tau degradation through the ubiquitin proteasome system, but this may not be reflected in the current assays that are specifically assessing oligomeric tau changes.

Response: To address a comment about “UBE4B/STUB1 still may mediate monomeric tau degradation through the ubiquitin proteasome system”, we updated Fig. 7a-c and 7f-l. These results were described in the result section of page 13-14 of the revised manuscript as seen below.

“In contrast, chloroquine significantly inhibited Tau degradation (Fig. 7b). In addition, pepstatin A (PEPA) alone and E64D plus PEPA (E64D+PEPA), autophagy inhibitors, significantly inhibited Tau degradation (Fig. 7c), suggesting that ALS was the major pathway of UBE4B and STUB1-mediated Tau degradation in neuroblastoma cells. Our *in vitro* study showed that the turnover of monomeric Tau molecule by UBE4B/STUB1 is preferentially mediated by autophagy-dependent manner rather than the proteasome pathway in SH-SY5Y cells (Fig. 7a-c).

Because autophagy inhibitor treatment blocked autophagic Tau degradation *in vitro*, we tested this in the *in vivo* system by injecting autophagy inhibitors into the dentate gyri of *Tau-BiFC* mice with co-expression of *UBE4B* and *STUB1* (Fig. 7d and e). We measured oligomeric Tau and phosphorylated Tau levels in the dentate gyrus of *Tau-BiFC* mice with the AT8 antibody (S202/T205) (Fig. 7f-h). Inhibition of ALS by Chloroquine and E64D+PEPA significantly increased oligomeric and phosphorylated Tau (S202/T205) (Fig. 7f-h). Similarly, the phosphorylated forms of Tau p-S396/S404 (PHF-1) and p-T231 (AT180) were also significantly increased in Chloroquine and E64D+PEPA-treated *Tau-BiFC* mice (Supplementary Fig. 7a-d). Since ALS disruption changes LC3 and p62 levels, which are inversely correlated with the efficiency of autophagy, we measured LC3 and p62 levels in our model system. Co-expression of *UBE4B* and *STUB1* significantly decreased LC3 and p62 levels relative to the control in the dentate gyri of *Tau-BiFC* mice, whereas autophagy inhibitors elevated LC3 and p62 levels relative to the control in the dentate gyri of *UBE4B+STUB1*-expressed *Tau-BiFC* mice (Fig. 7i and j). Furthermore, autophagy inhibitors modulated BECN1 levels compared to the control in the dentate gyri of *UBE4B+STUB1*-expressed *Tau-BiFC* mice (Supple Fig. 8). Collectively, these results suggested that monomeric and oligomeric Tau

degradations by STUB1 and UBE4B are mediated through the autophagy pathway rather than the ubiquitin proteasome system.”

b. In figure 6 (revised to Fig.7), authors ran an in-vivo chloroquine injection experiment to prove that UBE4B/STUB1-promoting degradation mediated autophagy. Authors need to inject chloroquine in the Tau-BiFC::UBE4B/STUB1 overexpressing mouse, not just Tau-BiFC mice to see if chloroquine is rescuing the Tau degradation facilitated by UBE4B/STUB1 overexpression.

Response: As suggested by reviewer 3, we prepared Tau-BiFC::UBE4B/STUB1 overexpressing mice and examined whether autophagy inhibitions (by CQ and E64D plus pepstatin A (E64D+PEPA)) block the Tau degradation by UBE4B/STUB1 overexpression. The immunofluorescence staining and its quantitation data with more autophagy markers such as LC3 and Beclin were added to Fig. 7i and k and Supplementary Fig. 8a and b. We have added this information in the result section of page 14 of the revised manuscript as seen below.

“Because autophagy inhibitor treatment blocked autophagic Tau degradation *in vitro*, we tested this in the *in vivo* system by injecting autophagy inhibitors into the dentate gyri of *Tau-BiFC* mice with co-expression of *UBE4B* and *STUB1* (Fig. 7d and e). We measured oligomeric Tau and phosphorylated Tau levels in the dentate gyrus of *Tau-BiFC* mice using the AT8 antibody (S202/T205) (Fig. 7f-h). Inhibition of ALS by Chloroquine and E64D+PEPA significantly increased oligomeric and phosphorylated Tau (S202/T205) (Fig. 7f-h). Similarly, the phosphorylated forms of Tau p-S396/S404 (PHF-1) and p-T231 (AT180) were also significantly increased in chloroquine and E64D+PEPA-treated *Tau-BiFC* mice (Supplementary Fig. 7a-d). Since ALS disruption changes LC3 and p62/SQSTM1 levels, which are inversely correlated with the efficiency of autophagy³², we measured LC3 and p62 levels in our model system. Co-expression of *UBE4B* and *STUB1* significantly decreased LC3 and p62 levels relative to the control in the dentate gyri of *Tau-BiFC* mice, whereas autophagy inhibitors elevated LC3 and p62 levels relative to the control in the dentate gyri of *UBE4B+STUB1*-expressed *Tau-BiFC* mice (Fig. 7i and j). Furthermore, autophagy inhibitors modulated BECN1 levels compared to the control in the dentate gyri of *UBE4B+STUB1*-expressed *Tau-BiFC* mice (Supplementary Fig. 8). Collectively, these results suggested that monomeric and oligomeric Tau degradations by STUB1 and UBE4B are mediated through the autophagy pathway rather than the ubiquitin proteasome system.”

c. Also, chloroquine administration is not sufficient to prove the autophagy mediation. It is true that chloroquine is known to inhibit autophagic activity, but its mechanism *in vivo* is not fully understood and other mechanisms, such as inflammatory modulation, are involved, which may also modulate tau pathology. The interpretation of this experiment should at least be softened.

Response: In order to address a concern about using one autophagy inhibitor, we further treated E64D+PEPA in Tau-BiFC::UBE4B/STUB1 overexpressing mice and examined whether the autophagy activity is important in the turnover of Tau molecule *in vivo*. We performed new immunofluorescence staining and analyzed image data with more markers of autophagy such as LC3 and Beclin. These results were added to Fig. 7i and k and Supple Fig. 8a and b. In addition, as suggested by reviewer 3, the interpretation of the results were added in the discussion section of page 18 of the revised manuscript as seen below.

“Considering autophagy inhibitors not only modulate neuronal autophagy pathway but also affect neuroinflammation pathway, it remains to be determined whether other neuroinflammatory pathways are involved in Tau pathology beyond the autophagy pathway in future studies.”

d. In addition, authors should measure a direct autophagosome formation marker (e.g. LC3), rather than only p62 which may cross into the UBS pathway.

Response: As suggested by reviewer 3, we performed immunofluorescence staining with more autophagy markers such as LC3 and Beclin. These new results were added to Fig. 7i and k and Supple Fig. 8a and b. We have added this information in the result section of page 14 of the revised manuscript as seen below.

“Since ALS disruption changes LC3 and p62/SQSTM1 levels, which are inversely correlated with the efficiency of autophagy³², we measured LC3 and p62 levels in our model system. Co-expression of *UBE4B* and *STUB1* significantly decreased LC3 and p62 levels relative to the control in the dentate gyri of *Tau-BiFC* mice, whereas autophagy inhibitors elevated LC3 and p62 levels relative to the control in the dentate gyri of *UBE4B+STUB1*-expressed *Tau-BiFC* mice (Fig. 7i and j). Furthermore, autophagy inhibitors modulated BECN1 levels compared to the control in the dentate gyri of *UBE4B+STUB1*-expressed *Tau-BiFC* mice (Supplementary Fig. 8).”

3. In general, the images provided in the manuscripts (in particular, in vivo studies) are not adequate and difficult to evaluate. For example, in the figure 5 (revised to Fig.6), it seems that Tau-BiFC is mostly expressed in the interneurons in the hilus of the DG. On the other hand, AT8 staining shows expression throughout the DG. This is a concern with the high concentration of primary antibody application (1:200). The authors need to show more controls for the AT8 staining.

Response: In order to address a concern about the image quality, we performed new immunofluorescence staining and image data analysis. These results were added in the revised Fig. 6c-g and 7f-l. In addition, the colocalization analysis in Supplementary Fig. 6d and e showed that Tau-BiFC (green) and AT8 (S202/T205) (purple) levels are correlatively reduced in the polymorphic layer of dentate gyrus in *AAV-UBE4B* and *AAV-STUB1*-dependent manners. BiFC signals is found in the dentate gyrus (Shin et al. *Pro. Neurobiol.* 2020 <https://doi.org/10.1016/j.pneurobio.2020.10178>) and AT8 is also found in the dentate gyrus and mossy fiber in the kainic acid-induced epilepsy mouse model (Alves et al. *Front. Aging Neurosci.*, 2019 <https://doi.org/10.3389/fnagi.2019.00308>). Taken together, the presence of Tau-BiFC and AT8 (S202/T205) signals in our AAV-delivered Tau-BiFC mouse model are well supported.

a. In addition, in the figures, the fluorescence intensities are different between supra- and infra-pyramidal blades in different groups (even in DAPI). This might be due to the variations in injection sites and area affected by virus transfections. The authors need to provide injection sites and spreading area including control groups.

Response: As suggested by reviewer 3, we added new image data to Fig. 6c that shows AAV injection foci and its expression pattern in the dentate gyri *in vivo*. We have added this information in the result section of page 12 of the revised manuscript as below.

“AAVs were delivered to the dentate gyri of *Tau-BiFC* mice by stereotaxic injection (Fig. 6b and c)”

b. Related to the issue above, authors need to provide the detailed experiment scheme with age and time line, to know what stage of tau pathology has been affected.

Response: As suggested by reviewer 3, we added experimental schema with age and time line in Fig. 6d and Fig. 7e. We have added this information in the result section of page 12 and 14 of the revised manuscript.

4. In figure 5 and 6 (revised to Fig.6 and 7), it was not clearly described how the number of subjects (n=4) and cell counting subjects {n=30 for figure 5 (revised fig.6) and n=20 for figure 6 (revised fig.7)} were generated. The cell counting/intensity of staining needs to be quantified and averaged as an individual subject level (n=4 per group) and treated as a sample for statistical analysis (instead of using multiple sections from an individual subject as separate n's).

Response: As recommended by reviewer 3, we rearranged the data to show the level of Tau as an individual subject level rather than a cell counting number in Fig. 6 and Fig. 7 as well as in Supplementary Fig. 6 and Supplementary Fig. 7. We have added this information in the figure legends.

5. The authors claim that the UBE4B and STUB1, but not alone, co-regulate tau degradation, in evidence by figure 4 (Tau was significantly increased by co-expression of UBE4B and STUB1, but not by expression of UBE4B or STUB1 alone). This argument is not consistent, necessarily, in the in-vivo study {figure 5 (revised to figure 6)}. Either UBE4B or STUB1 over-expression alone also reduced the tau levels. Can you authors explain this?

Response: As shown in Fig. 4b either STUB1 (lane f relative to lane a) or UBE4B (lane 1 relative to lane a) alone also triggers Tau degradation in SH-SY5Y neuroblastoma cells although to a lesser extent compared to the *in vivo* effect in Tau-BiFC mouse (Fig. 6 c and d). We believe this may be due to the difference between *in vitro* and *in vivo* studies. Overexpression on cells shows a very temporary and short-term effect. However, the *in vivo* system reflects the overall and long-term effects of living organisms. Thus, *in vivo* overexpression of either UBE4B or STUB1 can also interact with endogenous proteins to exert more physiological effects.

Minor

1. "Mohanty et al. (unpublished) demonstrated, line 320, that UBE4B ligase is also capable of both Lys48 and Lys63 polyubiquitination.", please discuss this more based on published data.

Response: We have added this information in the discussion section of page 18-19 of the revised manuscript as seen below.

“Lys63-linked polyubiquitination of Tau facilitates the formation of disease-associated Tau inclusions, which are preferentially cleared by the ALS, while Lys48-linked polyubiquitinated Tau is likely degraded by the UPS. Lys63 ubiquitinated substrates are recognized by autophagy receptors present on autophagosomes, mediating ALS-dependent degradation. Recent studies have shown that the knockdown of *UBE4B* affects Lys48 and Lys63 polyubiquitination in Tax binding proteins and Tax mediated activation of NF- κ B. Therefore, in the context of the present study, *UBE4B* acted synergistically with *STUB1* to facilitate Tau degradation by ALS, and could polyubiquitinate Tau proteins by Lys63 ubiquitin linkage.”

2. “The most significant reductions in eye sizes were induced by overexpression of miR-9a, miR-9b, and miR-9c”. miRNA- 989 and 932 also reduced eye size even more than miR-9a, miR-9b.

Response: We agree on this issue regarding the eye size of *miR-989* and *miR-932*. *Drosophila miR-989* has homology with mice and human *let-7* but they are not 100% homologous each other. *Drosophila miR-932* has no homology with mice and human *miRNAs*. Therefore, we chose *miR-9a* that shows 100% homology with mammalian *miR-9*.

3. Figure 4c. please report statistics using a repeated measure.

Response: We have corrected and added the explanation in the Figure 4c legend as below.

“Quantification of Tau levels was done considering the amount of β -actin protein in each case. Data represent the mean \pm SE of three independent experiments (* $P < 0.05$, ** $P < 0.005$, *** $P < 0.001$ two-tailed Student’s *t*-test).”

REVIEWERS' COMMENTS

Reviewer #1 (Remarks to the Author):

Accept

Reviewer #2 (Remarks to the Author):

The authors have adequately addressed all my previous concerns. In my opinion, the work is significant, appropriately documented, and suitable for publication.

Reviewer #3 (Remarks to the Author):

The authors have answered this reviewer's previous critiques very well. I have no further questions.